# Complex I is bypassed during high intensity exercise

Avlant Nilsson [1], Elias Björnson [1,2], Mikael Flockhart [3], Filip J. Larsen [3] & Jens Nielsen [1,4]*

Human muscles are tailored towards ATP synthesis. When exercising at high work rates muscles convert glucose to lactate, which is less nutrient efficient than respiration. There is hence a trade-off between endurance and power. Metabolic models have been developed to study how limited catalytic capacity of enzymes affects ATP synthesis. Here we integrate an enzyme-constrained metabolic model with proteomics data from muscle fibers. We find that ATP synthesis is constrained by several enzymes. A metabolic bypass of mitochondrial complex I is found to increase the ATP synthesis rate per gram of protein compared to full respiration. To test if this metabolic mode occurs in vivo, we conduct a high resolved incremental exercise tests for five subjects. Their gas exchange at different work rates is accurately reproduced by a whole-body metabolic model incorporating complex I bypass. The study therefore shows how proteome allocation influences metabolism during high intensity exercise.

[1] Department of Biology and Biological Engineering, Chalmers University of Technology, Gothenburg SE41296, Sweden. [2] Department of Molecular and Clinical Medicine/Wallenberg Laboratory, University of Gothenburg, Gothenburg, Sweden. [3] Åstrand Laboratory of Work Physiology, The Swedish School of Sport and Health Sciences, Stockholm, Sweden. [4] Novo Nordisk Foundation Center for Biosustainability, Technical University of Denmark, DK2800 Kongens Lyngby, Denmark. *email: nielsenj@chalmers.se

In the context of human endurance performance, high sustained ATP synthesis is one of the most important competitive advantages. It is increasingly recognized[1,2] that there is an evolutionary conserved trade-off between maximum-power output (using fermentative pathways) and maximum metabolic efficiency (using complete oxidative phosphorylation), which is rooted in differences in catalytic capacity of the different pathways (ATP produced per gram protein). The phenomenon is known as overflow metabolism. Human muscles can deploy the maximum-power strategy by producing lactate[3], but in contrast to microorganisms the human body is a closed system with regards to lactate and it must be metabolized elsewhere in the body under steady-state conditions.

A cells capacity to produce ATP depends on its protein composition. Microbes optimize their allocation of protein to their particular growth condition[4], enabling high specific growth rates. Similarly, physical exercise increases the levels of metabolic enzymes in human muscles[5]. However, few adjustments can be made to a muscle's protein composition during the course of an exercise bout, and it must be allocated so that it supports a wide range of ATP synthesis rates using different carbon sources, e.g., glycogen, glucose, and fat. The phenotypes of multicellular organisms do not only depend on the metabolic capacities of individual cells; while the maximal oxygen uptake rate in humans ($VO_2$max) is proportional to the oxidative capacity of the muscle fibers[6], it may also be limited by cardiopulmonary oxygen supply[7]. A number of potential limiting factors have been proposed, including the uptake capacity of oxygen by the lungs, the oxygen delivery due to the rate of blood flow, and the diffusion rate of oxygen into the cell[8]. To avoid overcapacity, most interacting components are expected to align, assuming that they are under selective pressure.

Cells must regulate the rate of ATP synthesis to match the demand. To avoid exhaustion of the ATP pool, ATP must be synthesized at the same rate as it is consumed, and metabolism therefore operates at steady-state during physical activity that lasts longer than a few minutes[9]. During exercise at varying work rates, the concentrations of key metabolites such as ATP and NADH barely change, despite large differences in ATP synthesis rates[9]. Some researchers argue that subtle changes in substrate concentrations are sufficient to regulate metabolism[10], others find them insufficient and propose that both supply and demand of ATP must be under the regulatory control[9]. Regardless of which regulatory mechanism may be at play, the metabolic states of the muscle are ultimately constrained by the catalytic capacity of the metabolic enzymes. The range of possible phenotypes that can be attained in muscle through regulation may therefore be investigated using constraint-based modeling.

Metabolic models are useful tools to study biological systems. They may largely be divided into kinetic models, which use ordinary differential equations, and stoichiometric models, which use linear equations and constraints[11]. Kinetic models are well suited to study the interactions between metabolite concentrations and fluxes and the roles of different regulatory elements. They have been used to study the biophysical mechanisms underlying experimental data in cells and isolated mitochondria, as well as the theoretical requirements on the regulatory system[9,10,12]. A kinetic model has also been developed on the whole-body level to simulate dynamics in substrate utilization[13]. Stoichiometric models are, on the other hand, well suited to study the metabolic steady states that cells may attain. They have been used to study the mechanisms behind overflow metabolism in microorganisms[14–17], as well as humans[3,18,19]. Unlike the microbial models that are constrained on the level of individual enzymes, the models[3,19] of human muscles have so far only been constrained by an estimated crowding constraint on the level of

each cellular compartment. Metabolic interactions between multiple tissues have also been studied using a stochiometric model[20], but was not applied to study fluxes during exercise.

We have previously shown that an enzyme-constrained model of yeast is able to capture shifts in metabolism across different metabolic regimes[14]. Here, we develop two metabolic models of human muscles, a small-scale enzyme-constrained model and a genome-scale, multi-tissue model. We use the small-scale model to simulate the optimal metabolic strategies at different ATP synthesis rates for different carbon sources. We find that metabolic bypass of complex I is an optimal strategy at high ATP synthesis rates. We integrate protein abundance measurements with the model to generate condition-specific constraints, and find that the capacity of complex I is constraining ATP synthesis. We incorporate these constraints into the multi-tissue model and fit the gas exchange of the model to experimental data from five subjects. A metabolic model without complex I bypass is unable to fit the fluxes. We use the model to study metabolic interactions between different muscle fiber types and peripheral tissue. We use the model to predict the maximum sustainable ATP synthesis rates at different exercise durations, and find that they correspond well to the data on world-record running speeds. Using these model systems, we were able to recapitulate known metabolic phenomena as well as predict previously unknown metabolic strategies in the working muscle during high-intensity exercise.

## Results

**There are four optimal metabolic strategies in muscle fibers.** Different metabolic strategies will be optimal depending on the type of physical activity, some strategies make efficient use of nutrients, whilst others favor rapid ATP synthesis. The optimal allocation of a finite pool of enzymes can be formulated as a linear optimization problem and studied using an enzyme-constrained model[14]. The model relies on estimating the mass of enzyme required for each biochemical reaction based on the specific activities of the enzymes (μmol [mg purified protein]$^{-1}$ min$^{-1}$). We reconstructed a small-scale model of intermediary metabolism (including 66 enzyme-mediated reactions) and parameterized it with specific activity values from literature (Supplementary Tables 1, 2). The model (Fig. 1a) covered all key ATP synthesizing reactions included glycolysis, TCA cycle, beta-oxidation, and oxidative phosphorylation (OXPHOS) and two routes for the oxidation of cytosolic NADH, the canonical malate–aspartate (mal-asp) shuttle and the glycerol phosphate (gly-phos) shuttle, which has previously been implicated in muscles[21]. Unlike the mal-asp shuttle which transports electrons from cytosolic to mitochondrial NADH, gly-phos transfers the electrons directly to ubiquinone ($QH_2$) that enter the electron transport chain at complex III. Fatty acids have a higher energy density than glycogen, and it has been hypothesized that muscles attempt to minimize glycogen utilization (glycogen sparing) by utilizing fatty acid synthesis when possible[22]. We configured the model to minimize the amount of carbon required for the synthesis of ATP (per cmol), to simulate the optimal utilization of nutrients.

The optimal enzyme allocation was simulated at different ATP synthesis rates. It showed four distinct metabolic modes that differed in catalytic capacity and substrate efficiency that together formed a Pareto front with the maximum attainable substrate efficiency at each ATP synthesis rate (Fig. 1b). A large share of the front consisted of mixtures of oxidative and fermentative ATP synthesis as is observed in human muscles. Interestingly, the gly-phos route for oxidizing cytosolic NADH, was predicted to be more catalytically efficient than the canonical mal-asp shuttle, which however, was more substrate efficient. Both the high

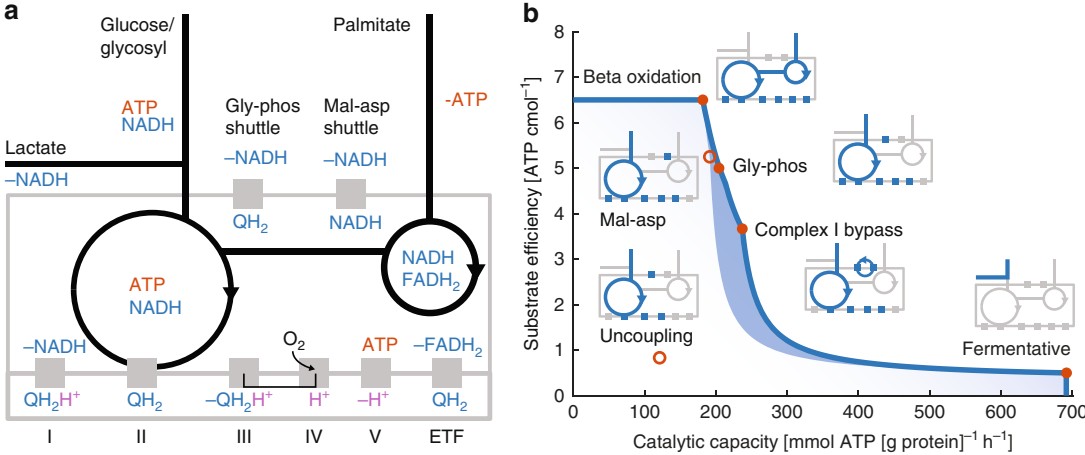

**Fig. 1** Bypass of complex I is a Pareto optimal metabolic mode. **a** Schematic representation of the model of intermediary metabolism, including beta-oxidation. **b** The model identifies four optimal (filled circles) metabolic modes with decreasing substrate efficiency (ATP per cmol), but increasing catalytic capacity (ATP[g protein]$^{-1}$ h$^{-1}$). These consisted of two fully oxidative modes on palmitate and glycogen, respectively, followed by a partially coupled oxidative mode (bypass of complex I), and a fully fermentative mode, two suboptimal metabolic modes are also shown for reference (open circle). The active pathways in each mode are indicated (in blue) in the schema corresponding to (**a**). Linear combinations of adjacent modes form a Pareto front for the trade-off between catalytic capacity and substrate efficiency. The mode including Complex I bypass increases the feasible area (indicated with blue shading), allowing a higher substrate efficiency than otherwise possible at high catalytic rates

substrate efficiency of oxidation and the high catalytic efficiency of fermentation were expected results that align with previous models and observations in microbes and human[3,16,18,19]. The concept of a partially coupled mode, through the bypass of complex I was less expected. However, this fits well with observations in rat[23], where differences in oxidative coupling, P/O ratio (ATP produced per molecular oxygen consumed) have been observed among different tissues respiring on substrates of complex I. It is also in agreement with the observed shift to isoforms of complex I with lower proton stoichiometry in *Escherichia coli* at high growth rates[24], and the absence of a proton-pumping complex I in *Saccharomyces cerevisiae*, as we noted in a previous study[14].

Since reducing the P/O ratio is expected to decrease the total amount of ATP synthesized, we were interested in how this became an optimal mode. We analyzed the model and found that the effect was driven by the low specific activity of complex I (Supplementary Table 3) emerging from its high-molecular mass, 980 kDa[25]. The main advantage over the fully oxidative mode is its high catalytic capacity; to synthesize equal amounts of ATP the fully oxidative mode would require a larger investment in proteins, with direct ATP costs for production and maintenance or opportunity costs in the form of reduced concentrations of other proteins if the total protein content is conserved. The main advantage over fermentation is the high substrate efficiency, which allows sustained ATP synthesis, additionally it does not result in the production of lactate, which cannot be sustained indefinitely. Complex I bypass thus enables a balance between ATP synthesis, protein allocation, and substrate efficiency, and constitutes a middle ground between the fully fermentative and the fully aerobic mode.

In the absence of complex I, mitochondrial NADH must be oxidized through a different route. We found that our model achieved this through transporting NADH to the cytosol via the reversed mal-asp shuttle and then using the gly-phos shuttle to transfers electrons from NADH to QH$_2$. While the gly-phos shuttle seems like a plausible mechanism for the oxidation of cytosolic NADH, it seems improbable that mitochondrial NADH would be routed to the cytosol, only to be taken up by the mitochondria again in the form of QH$_2$. A more sophisticated metabolic route for transferring electrons from NADH to QH$_2$

may exist in the living cell. One possibility may be to bypass the proton-pumping step of complex I, while retaining the NADH dehydrogenase activity. Such a switch has previously been proposed[26] and was recently experimentally demonstrated in *Yarrowia lipolytica*[27]. In *S. cerevisiae*, the enzyme NADH Dehydrogenase (NDI1) transfers electrons from mitochondrial NADH to QH$_2$ without pumping protons with a specific activity hundred times higher than for complex I[14]. To test if the model would favor a non-proton-pumping NADH Dehydrogenase, we temporarily added NDI to the model and found that complex I bypass was even more accentuated in the hybrid model (Supplementary Fig. 1). However, we were hesitant to introduce a hypothetical reaction into the model in the absence of direct evidence of a proton-slipping mode of complex I in human. For the rest of the simulations, the route over gly-phos was retained. This was not expected to constitute an active constrain in the rest of the analysis, since the specific activities of these reactions were comparably high.

**Muscle proteome allocation supports the four strategies**. Unlike microbes, human muscles cannot adjust their proteome during ATP synthesis. They must therefore be disposed to handle a range of different metabolic tasks. We evaluated if all four metabolic strategies would be reflected in the protein concentrations in muscle, as the protein concentrations together with their specific activities determine the flux capacities ($v_{max}$) of the different reactions. We investigated the published proteome of skeletal muscles (*vastus lateralis*) from eight healthy subjects (four males and four females)[28]. The proteome was dominated by motor proteins and metabolic enzymes (Fig. 2a), among the OXPHOS enzymes, complex V required more mass than the others combined, which is consistent with our analysis of yeast metabolism[14]. We then used the protein mass data to calculate the maximal flux capacity of each enzyme ($v_{max}$ = [protein concentration] × [specific activity]). The reactions were constrained to half max capacity to account for different effects, e.g., incomplete enzyme saturation, thermodynamically driven backward fluxes[29], effectively scaling the maximum flux to the expected biological range.

We simulated the optimal fluxes at various ATP synthesis rates using the constrained model. The carbon utilization was

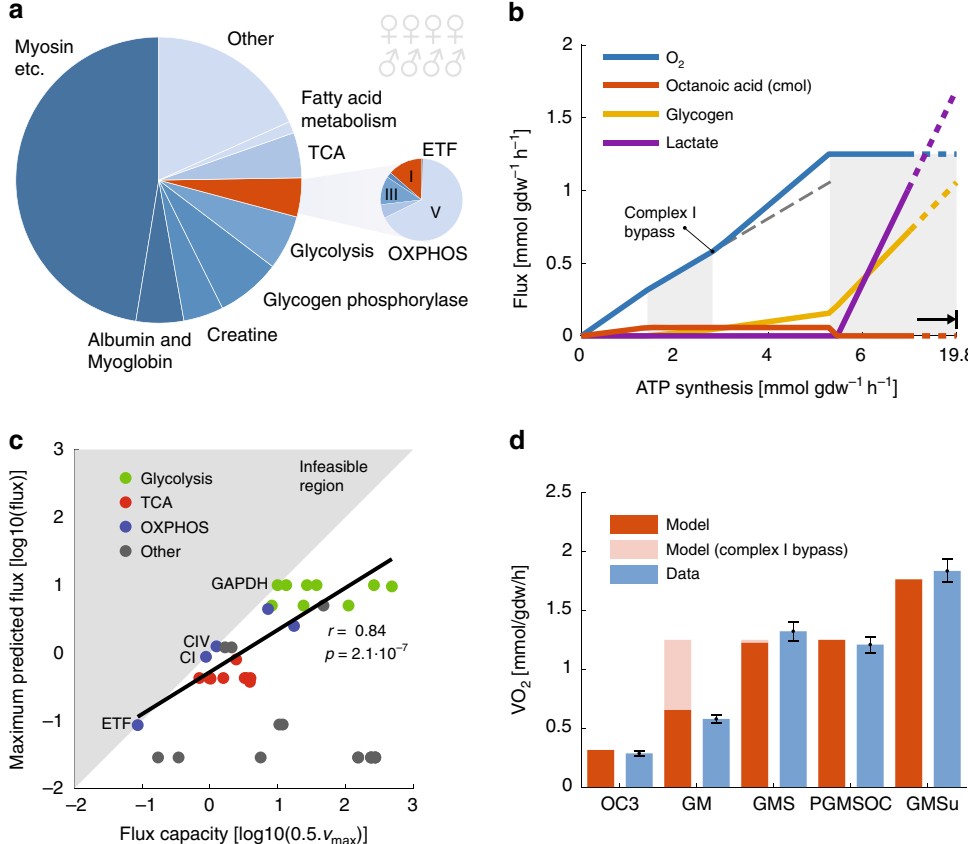

**Fig. 2** Metabolic modes in muscle fibers. **a** Most of the mass of muscle fibers, calculated from a published proteomics data set[28], is allocated to motor proteins and metabolic enzymes. **b** The predicted fluxes at increasing ATP synthesis rates show four linear modes (white and gray shading). Extrapolated oxygen flux (dashed gray line) highlights that the slope of $O_2$ increases after complex I bypass. All fluxes were calculated using enzyme concentrations as constraints and minimizing the substrate utilization (maximizing ATP per cmol). **c** The maximum predicted fluxes cannot exceed the flux capacity constraints (indicated by the shaded gray area). The predicted ATP synthesis is thus constrained by four enzymes in order, electron transfer flavoprotein dehydrogenase (ETF), complex I (CI), complex IV (CIV), and glyceraldehyde 3-phosphate dehydrogenase (GAPDH). There is a strong Pearson's correlation between flux capacity and predicted fluxes for the enzymes from glycolysis, citric acid cycle (TCA), and oxidative phosphorylation (OXPHOS). **d** The maximum predicted oxygen flux under different conditions was in good agreement with literature data under the corresponding conditions from muscle fibers[33]. The conditions were octanoic acid, an ETF substrate (OC); glutamate, a complex I substrate (GM); glutamate + succinate, a complex II substrate (GMS); glutamate + succinate + octanoic acid (PGMSOC3); glutamate + succinate in presence of a mitochondrial uncoupler (GMSu), this was simulated by removing constraints from the proton gradient-dependent OXPHOS enzymes. Each simulation was run with and without complex I bypass (blocked flux through gly-phos). The literature data were recalculated from wet to dry weight by an assumed water content of 80%, error bars show s.e.m.

minimized for each ATP synthesis rate, which is equivalent to maximizing the amount of ATP per carbon consumed. The simulations showed that the proteome of muscle is allocated in such a way that all four metabolic modes could be used, depending on the ATP synthesis rate (Fig. 2b). The predicted exchange fluxes agreed with known physiology[30], where metabolism is fully aerobic at low ATP synthesis rates, and fermentative at high rates. The maximum attainable ATP synthesis rate in the model was around 3.6 times higher than the rate at which oxygen uptake attained its maximum capacity. This is in good agreement with the relatively higher (3.2 times) maximum work rate on an exercise bike (1130 W) compared with the maximum aerobic steady-state rate (350 W)[31].

The capacity of the enzymes is expected to be dimensioned to align with the fluxes. A strong correlation (Pearson) was found between the flux capacity and maximum predicted fluxes among enzymes from glycolysis, TCA, and OXPHOS (Fig. 2c), suggesting that the protein composition of muscle is tailored toward ATP production. However, when also including fatty acid metabolism, the correlation vanished ($p = 0.88$). It is possible that enzymes from the beta-oxidation cycle require unusually high capacity to

overcome backward fluxes, since the cycle is found to be close to the equilibrium[32]. Fatty acid reactions aside, most reactions had an effective enzyme usage of between 10 and 100% (Supplementary Fig. 2). The enzyme usage of glycerol-3-phosphate dehydrogenase (GPD) from gly-phos was around 60%, confirming the assumption that it would not constitute an active constraint in the simulations.

Some metabolic fluxes were rate limiting in the model. It was possible (Fig. 2c) to directly identify which of the interior constraints were active by comparing the predicted fluxes with the flux constraints ($v_i = 0.5[v_{max}]_i$). By setting up simulations so that these constraints could not be overcome, it was possible to predict the maximum oxygen uptake rates in different metabolic context that were in good agreement with the corresponding literature data[33] from permeabilized muscle fibers (Fig. 2d). The much higher oxygen flux observed in the presence of complex II is in favor of a capacity constraint in complex I, and assumes that complex I bypass is not induced in the muscle fibers, e.g., directly by the complex I substrates. This type of the experiment is commonly performed with carbon sources that are optimized for the assay and not necessarily physiological relevant, e.g.,

octanoate instead of palmitate and glutamate instead of glucose. We therefore selected the carbon source in our simulations to mimic the experimental setup.

**The oxygen usage per watt increases at high work rates**. We were interested in if complex I bypass could affect whole-body metabolism during exercise. The metabolic rate of muscles can be experimentally controlled during physical activity on an exercise bike (ergometer) by adjusting the resistance[34]. The efficiency between power output measured on the ergometer and ATP production in the cell is expected to be ~35–50% based on experiments and biophysical simulations[35–38], which is in good agreement with an experimentally estimated output of 27 J per mmol ATP found for active muscle performing high-intensity aerobic exercise[39].

A bypass of complex I is expected to increase the oxygen expenditure at high work rates. Compared with the fully aerobic mode at low work rates, the oxygen expenditure per watt produced at high work rates is expected to increase up to 43%, assuming that all muscle fibers bypass complex I simultaneously and that glycogen is the sole carbon source. In the literature data[40], the median increase in oxygen expenditure is ~30% (Fig. 3a). But from these experiments, it is not possible to assess if attenuating factors, such as an accumulation of lactate or a switch in carbon source at high work rates also may be influential.

To be able to study these processes closer, we measured the gas-exchange fluxes in a cohort of five well-trained subjects (Supplementary Data 1). The increase in oxygen expenditure was similar to the increase found in the literature (Fig. 3b). The fine-grained incremental exercise protocol was designed to be able to pinpoint ($CI_{max}$), where the excess increase in oxygen expenditure would begin (Supplementary Fig. 3a). For most subjects, this occurred at ~40% of $VO_2$max (Fig. 3c), which was consistent with 42% for the switch in the simulated muscle (Fig. 2b). It is possible to quantify which carbon source is used through indirect calorimetry. The respiratory exchange ratio (RER) is the quotient of $CO_2$ over $O_2$, the value depends on if fat (RER = 0.7) or glycogen (RER = 1) or a mixture (0.7 < RER < 1) is used as carbon source. The exchange of $CO_2$ and $O_2$ were quantified, and the RER was significantly higher at high work rates (Supplementary Fig. 3b), consistent with an attenuated increase in $VO_2$ through shifts in carbon source.

**Metabolic strategies in a multi-tissue model**. A number of metabolic factors influence the oxygen expenditure at the whole-body level. The rate depends on the carbon source as well as the mixture of oxidative and fermentative ATP. Prolonged lactate production at high work rates is unsustainable, it is, however, actively metabolized during exercises[41], primary to $CO_2$[42], i.e., for ATP synthesis. At low exercise rates, only the oxidative type I fibers are activated, while type II fibers are found to activate at ~40% of $VO_2$max[43]. To take all of these factors into account, we constructed a multi-tissue model of the human body (Fig. 4a) that was parameterized (Supplementary Table 4) to fit the pulmonary exchange fluxes of $CO_2$ and $O_2$ for each of the subjects. The maximum uptake of $O_2$ was constrained to the observed $VO_2$max of the subject, while $CO_2$ as well as palmitate and glycogen were allowed to exchange freely over the boundary. To be able to accurately fit the $CO_2$ fluxes and thus the RER values, the capacity for fatty acid utilization (the ETF constraint) was fitted for each subject. Beyond a critical work rate, the RER exceeds 1, which is driven by the release of $CO_2$ from bicarbonate ($HCO_3^-$) due to buffering of lactate[44]. Under steady-state conditions, this effect is zero, but during lactate accumulation there is a 1:1 relation between the $CO_2$ and the lactate flux. To accommodate RER

values exceeding 1, we allowed a manually fitted lactate accumulation flux.

The predictions from the multi-tissue model were in good agreement with experimental measurements. Optimal fluxes were simulated at increasing work rates in the multi-tissue model. For each ATP synthesis rate, the use of glycogen and oxygen as well as production of lactate was minimized. The simulated exchange fluxes of $CO_2$ and $O_2$ as well as the RER fitted the experimental data well (Fig. 4b that shows the results for one representative subject, see Supplementary Fig. 4a for the remaining four subjects). The capacity constraint on complex I was crucial for the model to fit the fluxes, also the best-fit parametrization of the model without complex I bypass failed to describe the experimental data (Supplementary Fig. 4b). The oxygen usage by the model was attenuated by a shift towards glycogen as carbon source, which is more oxygen efficient than fat (Fig. 4c). The experimentally determined shift in RER was consistent with a saturated fat consumption rate occurring at ~100 W, diluted by increasing glycogen usage (Supplementary Fig. 6, and simulated fluxes in Supplementary Data 2). Much of the variance in $dVO_2$ per dW observed for different subjects (Fig. 3a) could be explained (Supplementary Fig. 5) by perturbing parameters in the model. This confirmed that complex I bypass is a plausible cause of the increased oxygen expenditure at high work rates.

At the highest work rates, the model predicted the excretion of lactate by m2 (type II fibers) and metabolism of lactate by m1 (type I fibers). This occurred when complex IV attained its maximum capacity in m2; it then became optimal for the model to abandon fatty acid consumption and replace it with glycogen consumption (since only glycogen can be fermented). Simultaneously, m1 was forced to switch from palmitate as a carbon source, to act as a sink for the excess lactate. Since our model assumes steady-state, it cannot capture dynamic effects when lactate begins to accumulate. This may explain why our predicted RER values are lower than observed around the lactate threshold. This may also explain the slow component phenomenon[45] that involves a slow increase in oxygen consumption at high work rates over several minutes until the steady-state oxygen consumption is reached. It has previously been linked to lactate dynamics, since the effect disappears in the presence of saturating lactate concentrations. It was indeed possible to explain the literature data[45] using a simple dynamic model (Supplementary Fig. 7) that described the shift in carbon source from glycogen to lactate as a response to increased lactate flux from type II to type I muscle fibers.

**The model is predictive of world-record running speeds**. The maximum work rate of a muscle depends on the carbon source. We used the model to investigate the maximum steady-state ATP synthesis rate on different carbon sources (Fig. 5a). The rate on intracellular glycogen was higher than on glucose, i.e., liver glycogen converted to glucose and released into blood, due to the lower ATP yield from glycolysis on glucose (two ATPs per glucose) compared with glycogen (three ATPs per glycogen), arising from the additional ATP consumption by hexokinase compared with glycogen phosphorylase. This under the assumption that glycogen is being depleted, as re-synthesis of glycogen requires one ATP per glucose. Under fermentative conditions, this results in 50% more lactate produced per ATP synthesized, leading to saturating lactate fluxes at lower work rates. The lower work rate on lactate and fat was was due to the lower P/O ratio and lack of fermentative capacity. For fat, we noticed a marked discrepancy in uptake rate if ETF was unconstrained (~1.8 g per min) compared with when it was empirically constrained to fit the RER values (~0.6 g per min). The low empirical utilization of fat may

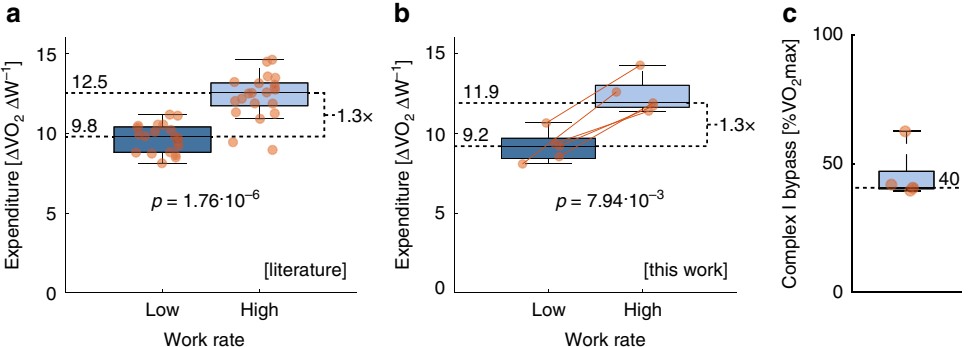

**Fig. 3** Metabolic modes in subjects on exercise bike. **a** The oxygen expenditure, oxygen consumed ($VO_2$) in ml per min per watt produced (W), is higher at high work rates for subjects ($N = 21$) on an exercise bike, as calculated from published data[40]. **b** This was reproduced for the $n = 5$ subjects studied in this work. **c** The $VO_2$ when the derivate increases for each subject was consistently calculate to ~40% of the maximum $O_2$ flux. Statistics were calculated using two-sided Mann–Whitney U test. The box plot indicates data between the 25th and 75th percentiles, the central mark indicates the median and whiskers extend to the most extreme data point within $1.5 \times$ interquartile range

**Fig. 4** The effects of complex I bypass in a multi-tissue model. **a** Schematic representation of the multi-tissue model. Three genome-scale models corresponding to muscle fibers with high mitochondrial content (type I fibers, m1), intermediary mitochondrial content (type II fibers, m2), and peripheral tissues (m3) were joined by a shared extracellular compartment (b) resembling transport through the blood stream. The flux through ETF, CI, and CIV was constrained in m1 and m2, and scaled by a constant to allow for differences in enzyme concentrations between subjects. **b** It was possible to fit the model (orange line) to experimentally observed $VO_2$ and $VCO_2$ data (blue circles) from a subject performing incremental exercise on a bike. The lactate concentrations in blood, which was obtained for the same subject at a different occasion, could also be fitted assuming a Michaelis–Menten relation between the concentration and the predicted lactate flux. The fit for the model with complex I bypass was better than for a model without (gray line). The oxygen expenditure was highest in a region (indicated with gray shading) between complex I bypass and before lactate accumulation, as given by RER > 1 (dashed line). **c** Relative substrate utilization in the two muscle fiber sub-models (cmol) showed a shift from fatty acids as carbon source towards glycogen. After complex IV was saturated in m2 the model predicted a flux of lactate from m2 to m1

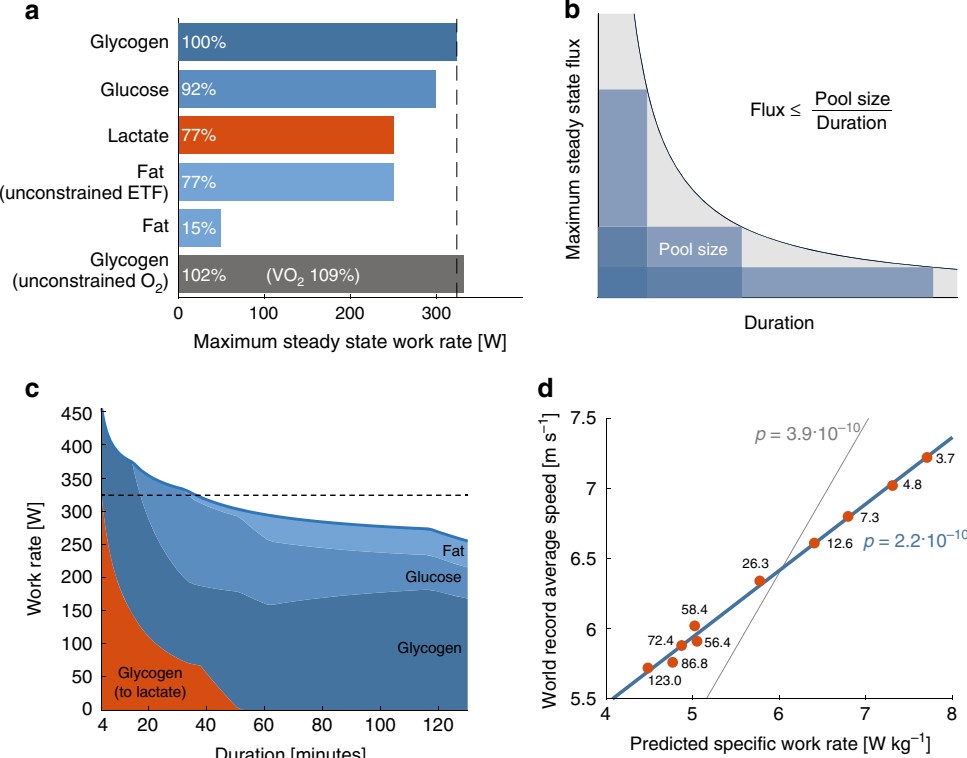

**Fig. 5** Maximum work capacity on different substrates and at different durations. **a** There are differences in the maximum predicted work rate for different carbon sources, assuming unconstrained uptake. **b** The maximum steady-state flux is constrained by exercise duration, assuming substrates are depleted at the end of the exercise. **c** The maximum attainable work rate for a given exercise duration with maximum steady-state work rate (dashed line) for the reference. The colored regions show the relative flux of carbon on a carbon mol basis. **d** The decrease in running speed at increasing distances may largely be explained by limited metabolite pool sizes. There is a linear relation between the predicted maximum specific work rate (predicted work rate per kg bodyweight) at different durations and the world-record average running speeds at these durations (3.7–123 min), regression without intercept (gray line) and with intercept (blue line), $p$-value from the $t$-statistic of the slope

suggest a proteomic adaptation to the lower catalytic efficiency of fat (Fig. 1b) and low capacity for fat oxidation in muscle fibers (Fig. 2d).

The steady-state solutions identified by the model become invalid if substrate pools are depleted (Fig. 5b). Glycogen reserves are a known limiting factor for athletes exercising for long durations, while for shorter durations the accumulation of lactate is limiting[46]. We simulated the maximum work rate that could be sustained for different durations of exercises by constraining the uptake rate of different carbon sources by the pool size divided by the duration of the exercise. For lactate, we assumed that it accumulated in blood and tissues (a total volume of around 37 l) with a final lactate concentration of around 15 mmol per l[43], allowing high lactate efflux at short durations. With these constraints, we arrived at a map of the metabolic capacities in exercise bouts of different durations (Fig. 5c). At short durations, a large fraction of the glycogen was diverted toward lactate production, while fluxes are high, the increase in ATP synthesis is modest, due to the order of magnitude lower ATP synthesis rates for fermentation compared with OXPHOS. At durations above 20 min, glycogen reserves become limiting and other carbon sources were introduced and at durations above 40 min the carbon resources became too scarce to justify lactate accumulation.

To verify our predictions, we compared them with the world-record running speeds. The average running speed in exercising humans decreases as a function of the distance (and hence duration) of the race, which is related to the decreasing power output[47]. Professional athletes are highly motivated and are expected to perform near optimally, which has previously been exploited by mathematical models that predict world-record performance[47]. We compared the current world-record running speeds at various durations[48] with the predicted optimal-specific power output of one of our subjects for the same durations (Fig. 5d) and found a close linear relationship ($R^2$ 0.995). This corroborates our model predictions and suggests that much of the decrease in running speed at longer distances may be understood from the optimal utilization of different carbon sources with limited pool sizes. Other factors, including air drag, and differences in protein allocation and body composition between runners are expected to attenuate the effect, and may explain the larger dynamic range for the predicted values compared with the observed.

## Discussion

We investigated the metabolic fluxes in muscle fibers at different work rates using metabolic modeling. A metabolic model constrained with protein concentrations from muscle predicted that complex I is bypassed at high ATP synthesis rates. The oxygen fluxes predicted by the model were in good agreement with the literature data for muscle fibers. The bypass of complex I is predicted to increase the catalytic capacity of the muscle at the expense of substrate efficiency, an effect that emerges from the low specific activity of complex I. While the model predicted that complex I is bypassed through the well-established[21] gly-phos shuttle, it is, however, not impossible that it is bypassed through another mechanism in vivo, e.g., disengagement of the proton

pumps in complex I[27]. This seems particularly probable for NADH of mitochondrial origin. Regardless of mechanism, a whole-body model that included complex I bypass showed a good fit to exchange fluxes of $O_2$ and $CO_2$ from human subjects, while the fit for a model without complex I bypass was poor. Furthermore, taking the pool sizes of different carbon sources into account, the model was able to predict a decreasing trend in power output with increasing durations, that was consistent with the trend for world-record running speeds.

To maximize performance, the muscle must allocate the protein composition for the tasks it is likely to encounter. A coarse-grained adaptation appears to occur through the composition of muscle fibers, by tuning the ratio of glycolytic type II fibers and the more oxidative type I fibers. But the body is also capable of adjusting the catalytic capacity of individual fibers[5]. From this perspective, it is interesting to note how different forms of doping may improve performance, e.g., the maximum work rate and $VO_2$max can be increased in response to drug erythropoeitin EPO, which is intended to increase oxygen delivery. In a recent randomized controlled trial[49], the placebo group experienced a 2% increase in both maximum work rate and $VO_2$max (12.5 $dVO_2$ per dW), while the increase was 5% in maximum work rate and 9% in $VO_2$max for the EPO-group (20.8 dVO2 per dW). Again suggesting that a linear relation between oxygen expenditure and work rate may be a too simple model.

Regarding optimal substrate utilization, the model predicts a switch from mixed oxidation of fatty acids, glucose, and glycogen to exclusive glycogen oxidation at high work rates. This is due to the extra ATP gained from glycolysis compared with beta-oxidation. This is in agreements with dynamic models of muscle tissues that predicts that most (70–90%) of ATP originates from carbohydrates at high work rates, out of which most (~80%) originate from muscle glycogen[13], as well as experimental values at ~66% and 70%, respectivly[50]. The low empirical utilization of fat as a substrate may be due to an impaired ability to deliver fatty acids to the muscle, as was previously proposed[50]. The composition may also reflect an adaptation to a high carbohydrate diet, as the utilization of fat is shown to increase on a high-fat diet[51], however, this also reduced their maximum work rate. Furthermore, postprandial effects may influence the disposition toward carbohydrate metabolism. It is also interesting to notice that glucose is predicted to be a suboptimal fuel compared with glycogen since glycogen bypasses the first ATP-demanding step of glycolysis. This emphasizes the value of glycogen loading[52] in favor of carbohydrates from exogenous sources. The later should, however, be optimal at longer distances, as the uptake rate has been observed at a maximum of 100 g per h using a mixture of glucose and fructose[53]. At high work rates, lactate produced by one muscle fiber may be consumed by another, allowing a division of labor, but since lactate is a less oxygen efficient substrate than glycogen, it will increase the overall oxygen expenditure by the muscle. At the highest work rates, lactate accumulates, and the associated physiological disturbance quickly results in exhaustion.

Complex I bypass may play an important role in many pathological and paraphysiological conditions. Complex I is implicated in tumorigenesis, but there are conflicting observations with regards to its role. Many studies suggest that several subunits of complex I act as tumor suppressors and that mutations promote cancer progression, yet inhibition of complex I is the proposed mechanism of action for many anticancer compounds[54]. This conflict may potentially be resolved by the concept of complex I bypass, assuming that it occurs through the disengagement of the proton pumps; mutations may induce complex I bypass and increase the catalytic activity of the cancer, increasing its proliferation, inhibitors of complex I may on the other hand also inhibit the NADH to QH₂ reaction, effectively blocking complex I bypass. The relative expression of glycolytic and OXPHOS proteins shift under many paraphysiological conditions[55,56]. Here, complex I bypass may play an important role, given its intermediary location between oxidative and fermentative metabolism (Supplementary Discussion).

While the model has demonstrated merits in predicting phenotypes and providing systems-level insight, the phenomena that have been described herein, e.g., complex I bypass, require additional experiments and dynamic modeling to elucidate their mechanisms and regulation. Steady-state modeling is a useful paradigm, but many biological phenotypes are transient or rely on finite resources, e.g., the transient peaks in lactate concentration observed during moderate exercise[57]. We have been able to overcome these limitations to some degree through the use of duration-dependent uptake rates and a simple dynamic model. But most of the results depend on the steady-state assumption. In our simulations, we also assume that all enzymes operate at half their maximum rate, which is an apparent simplification and new types of models are under development that will allow the saturation of each enzyme to be estimated[58]. Nevertheless, steady-state modeling of metabolism may, thanks to the low number of parameters required and its intuitive application, serve as a framework to further our understanding on metabolic adaptations to exercise and limitations to human performance.

In conclusion, the model developed here recapitulates long known metabolic strategies during high-intensity exercise, such as lactate production and a switch toward glycogen as carbon source. It also makes several predictions of metabolic strategies during high-intensity exercise, including complex I bypass and lactate accumulation. These metabolic strategies explain several observations relating to altered oxygen expenditure during exercise.

## Methods

**Genome-scale model.** A genome-scale metabolic model of the muscle[59] was used with some additional manual curation (as specified in Supplementary Methods). The model originates from a generic GEM[60] that has been reduced to a muscle model by removing reactions without experimental support in the proteomics and transcriptomics data. Simulations using flux balance analysis (FBA) were carried out using the RAVEN Toolbox[61].

**Reduced muscle model.** A reduced stoichiometric model was constructed by simulating ATP production in the genome-scale model and removing the reactions that did not carry flux. To ensure that multiple metabolic strategies would be available also in the reduced model, multiple simulations were conducted where flux through important enzymes were blocked, e.g., enzymes from the respiration chain (Supplementary Methods).

**Enzyme-constrained model.** The reduced model was expanded to an enzyme-constrained model. Enzyme-specific activity values were collected for each metabolic reaction from an online database[62] and literature (Supplementary Tables 1, 2). An assumed saturation of 0.5 was used to calculate the estimated sum of enzyme mass required for each flux distribution. The sum of mass was then constrained to a finite value. By increasing the ATP synthesis rate in the simulations, the catalytic capacity and substrate efficiency can be calculated (pseudo code in Supplementary Methods).

**Proteomics analysis.** The mean spectral count from eight subjects was calculated for each protein in a published data set[28]. The mass fraction of each protein was calculated by multiplying the average spectral count by the molecular weight of each protein and dividing by the sum. The mass fraction was then converted to absolute protein abundance through multiplication with the protein content of muscle[63]. Proteins were binned to metabolic processes based on the gene annotation in the GEM. Proteins without annotation were manually binned to the category "myosin" (containing myosin, actin, and similar proteins) or binned as "other". Myoglobin and albumin were given their own category due to their high abundance.

**Proteome-constrained simulations.** The $v_{max}$ of each enzyme-catalyzed reaction of the reduced model was calculated from the relationship $v_{max}$ = [protein mass concentration] × [specific activity]. The flux of each reaction was constrained to an

estimated effective $v_{max}$ as $v \leq \eta \times v_{max}$, were the factor $\eta$ ($0 \leq \eta \leq 1$), which represents incomplete enzyme usage, was taken as $\eta = 0.5$ for all reactions. The uptake of oxygen, glycogen, and octanoic acid was left unconstrained. The ATP synthesis rate was fixed at incrementally increasing values, and the optimal flux distribution was calculated using FBA by minimizing the influx of carbon, subject to the effective $v_{max}$ constraints. To simulate oxygen expenditure in muscle fibers in different metabolic contexts, the ATP synthesis was maximized. A reaction that provided glutamate and removed aspartate was added to simulate the complex I substrates, and a reaction that provided succinate and removed fumarate was added to simulate complex II substrates. To simulate the oxygen flux in absence of complex I bypass, the gly-phos reaction was blocked. For the enzymes TPI and PFK, no measurements were found in the proteomics data, and these reactions were therefore not constrained. For TPI, the specific activity was sufficiently high to not be limiting even if protein levels were below the limit of detection. For PFK, there was a "PFK like protein" detected, but this was not used in the simulations (Pseudo code in Supplementary Methods).

**dVO$_2$ per dW calculations from literature**. The dVO$_2$ per dW metric at high and low work rates was calculated from a published data set (21 subjects) from an incremental exercise protocol on bike[40]. For low work rates (work rates below the lactate threshold), the reported slope of regression was used. No slope was reported for high work rates so there dVO$_2$ per dW was calculated from the difference in reported VO$_2$ and W at the lactate threshold (LT) and at max. The p-value was calculated using Wilcoxon rank-sum test.

**Incremental exercise measurements**. The experiments on human subjects were approved by the Swedish Ethical Review Authority and conformed to the declaration of Helsinki. Five male subjects (age $38 \pm 5$ y, height $181 \pm 3$ cm, weight $76 \pm 11$ kg, VO$_2$max $62 \pm 11$ ml kg$^{-1}$min$^{-1}$, W$_{max}$ $5.2 \pm 1$ W per kg, $\pm$ s.d.) completed a high-resolution incremental test to measure VO$_2$ and CO$_2$ gas exchange in expired air at a variety of work rates. The subjects were all experienced cyclists, and refrained from the training (36 h), caffeine (6 h), and food (3 h) preceding test and received information and gave written consent before test. All cycling during the submaximal test was performed at 70 rpm on an ergometer (SRM international, Jülich Germany) with continuously breath by breath measurement of VO$_2$ and CO$_2$ (Oxycon Pro, Erich Jaeger GmbH, Hoechberg, Germany). Due to technical limitations of the bike and the need for steady-state O$_2$ consumption, low work rates (rest to 100 W) was measured at durations of 2–5 min at increasing work rates of 10–20 W per step. At higher work rates (80 W up to a power output eliciting ≤ 1.00 RER), the work rate was increased in steps of 5 W per 90 s. The power- and gas-exchange data were time aligned and 15 s means were calculated. After the submaximal test, a short incremental test to exhaustion was performed for determining VO$_2$max. On a separate occasion lactate was sampled (EKF-diagnostic GmbH, Germany) at 5 min long intervals for one of the subjects during a standard LT-test with increments of 30 W (rest to 330 W) used at the laboratory. It is assumed that the experimentally determined fluxes were at steady-state, since the increments in work rate were small and each work rate target was maintained for a long time.

**dVO$_2$ per dW calculations in this study**. The point at which the derivative of VO$_2$ changed and its 95% confidence interval was calculated using nonlinear least squares curve fitting on the experimental data from five subjects. The slopes before and after the point were calculated using linear regression.

**Multi-tissue model**. Three genome-scale models of the muscle were combined to a multi-tissue model by a transport compartment that joined the exchange fluxes from all sub-models, allowing exchange of metabolites between the sub-models (Supplementary Fig. 7). The transport compartment was not constrained, under the assumption that blood flow is adjusted during exercise to support the fluxes. To represent the lower oxidative capacity of type II fibers, their capacity constraints were reduced to 50% of the values in type I fibers. An ATP hydrolysis reaction was added to each model. Maintenance energy expenditure was simulated by forcing ATP hydrolysis in m3, fitted for each subject, to account for metabolic activities that do not take place in the active muscle. Internal muscle work against gravity was simulating by forcing ATP hydrolysis in m1 at a rate of 2 mol ATP per h. To represent the preference for ATP synthesis by type I fibers at low work rates, they were given a positive weight in the objective function, but the rate at which ATP could be produced by type I fibers alone was constrained, and above this rate ATP synthesis was allowed from both muscle fibers in a fixed ratio. Lactate efflux was permitted from m2 to b and lactate uptake was permitted from b to m1 (construction of the multi-tissue model further specified in Supplementary Methods). A conversion factor is required to relate the power output of the ergometer to the ATP production by the model. The thermodynamic efficiency of muscle motor enzymes (myosin) is consistently found between 35 and 50% in experiments and biophysical simulations[35–38], which provides a conversion factor of 21–30 mmol ATP per s to W (or equivalently J [mmol ATP]$^{-1}$) assuming a $\Delta G$ of 60 KJ per mol ATP[37]. This is in good agreement with an experimentally estimated value of 27 J per mmol ATP found for active muscle performing high-intensity aerobic

exercise[39], which was used for the simulation. The gas exchange was converted assuming a concentration of 44.66 mmol oxygen per liter.

**Simulation of incremental exercise**. An objective function was devised to capture the behavior of muscle, minimizing the usage of oxygen, glycogen and the accumulation of lactate. The uptake of glycogen and fat was set unconstrained, and the objective function was minimized for each investigated work rate using FBA (pseudo code in Supplementary Methods).

**Simulation of optimal steady-state work rate**. The accumulation of lactate and uptake of each substrate was constrained to 0. The uptake of each carbon source was then permitted in turn, and the ATP synthesis was maximized using FBA. With lactate and fat as carbon sources, the peripheral tissue was allowed to consume the substrate to support maintenance (pseudo code in Supplementary Methods).

**Simulation of optimal work rate at increasing durations**. The reserves of glycogen of ~80 mmol per kg wet weight[64] for the m1 and m2 tissue were divided by the duration of the exercise to calculate the maximum steady-state influx at each duration. Glucose flux was calculated from an assumed liver glycogen store of 100 g, corresponding to 0.56 mol. Fat uptake was left unconstrained. The lactate concentration in blood was constrained to not exceed 15 mmol[43]. For each investigated duration, the work rate was maximized using FBA (pseudo code in Supplementary Methods).

**Reporting summary**. Further information on research design is available in the Nature Research Reporting Summary linked to this article.

## Data availability
The authors declare that the data supporting the findings of this study are available within the paper and its Supplementary information files.

## Code availability
The code was implemented in Matlab (Matlab_R_2018_b). Models and source code for the computational methods are made available through an online repository (https://github.com/SysBioChalmers/MuscleATPProductionSimulation).

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

## Acknowledgements

We acknowledge funding from the Novo Nordisk Foundation, Knut and Alice Wallenberg foundation, the ERASysAPP project IMOMESIC, funded by the The Swedish Research Council and Västra Götaland Regional Council, and Swedish Research Council for Sport Science. A.N. and E.B. conceived and initiated the project. A.N. constructed the model, ran the simulations and analyzed the model results. E.B. supported A.N. in the structure of the modeling work and analysis of results. A.N. wrote the paper with support from E.B. F.L. and M.F. performed in vivo experiments, analyzed the data, and reviewed the paper. J.N. supervised the work, provided funding, and edited the paper. Open access funding provided by Chalmers University of Technology.

## Competing interests

The authors declare no competing interests.
