## [Peer Review File · Nature Communications]

Reviewers' Comments:

Reviewer #1:

Remarks to the Author:

This study used an enzyme constrained metabolic model to examine how much ATP can be produced by bypassing Complex 1 of the electron transport chain (ETC). It is suggested that the model developed recapitulates long known metabolic strategies during high intensity exercise, such as lactate production and a switch towards glycogen as a sole fuel. It also predicts a number of metabolic changes that occur during exercise, which includes Complex 1 bypass, mitochondrial uncoupling and fatty acid synthesis in peripheral tissue.

General Comments:

The concept of a ETC complex 1 bypass is interesting during intense exercise is interesting, but I have some concerns with the metabolic model that the authors used. It is not described in detail, and appears to be heavily "glucose" centric (see Figure 1A). I am not sure that the contribution of other carbon substrates, particularly fatty acids, is adequately considered in the model. This is particularly relevant to the proposed conclusion of a Complex 1 bypass, as much of the FADH₂ entering the ETC via Complex 2 originates from fatty acid β -oxidation. It is also not considered that during exercise circulating fatty acid levels increase, and can also provide an additional source of fuel for muscle.

My major concern with the Complex 1 bypass model is where all the NADH is going. The major reduced equivalents produced is NADH, not FADH₂, and if Complex 1 is bypassed then muscle would need to deal with all this NADH. The authors propose that this NADH can be rerouted through the glycerol phosphate shuttle. Due to the amounts of NADH produced I find this hard to believe, and the authors do not provide any convincing evidence to support this possibility.

The concept is proposed that bypassing Complex 1 is a biological adaptation of muscle to situations such as exercise. However, it is not clear what advantage this bypass of Complex 1 would have during exercise. The net effect would be a decreased ATP production.

The authors propose that glycogen use provides additional ATP (3 ATP/glucose) compared to glycolysis (2 ATP/glucose). This is actually only relevant in a net glycogen depletion scenario. The synthesis of glycogen requires the consumption of 1ATP/glucose).

The conclusion that metabolic changes during exercise result in mitochondrial uncoupling and fatty acid synthesis in peripheral tissue is speculative and not based on evidence or modeling.

Figure 2 C predicts that fatty acids are minor contributors to metabolic flux. This is hard to believe and not consistent with actual metabolic measurements of fatty acid flux. Indeed, in Figure 1B, the RER values are consistent with a significant contribution of fatty acid oxidation to oxygen consumption.

I have concerns that the experimental validation of the modeling was performed using a a high resolved incremental exercise test on only one individual.

Reviewer #2:

Remarks to the Author:

The major claims of the paper are:

- 1)the ATP synthesis is strictly regulated by the specific activity of the metabolic enzymes
- 2)The amount of ATP that can be produced per gram of protein can be increased by bypassing complex I

3) A third mode to produce energy exist by bypassing complex I.

The observation made are novel and I believe the paper is of interest for the scientific community , not only in endurance training but particularly on paraphysiological and pathological conditions such as adaptation to hypoxia and muscle disorders, in which it is known that the metabolic adaptation is one of the primary events to cope with muscle impairment.

However there is an aspect that is not well explained, even if I understand that this was not probably relevant for the Authors. There is not a proper description of the type of muscle utilized to calculate in 8 healthy subject the absolute amount of proteins. Muscles are characterized by different fiber composition and the variability among subjects could be quite high, I think that a brief description could be helpful to better appreciate the results. In addition there are not reference of papers describing the metabolic rewiring revealed by semiquantitative data obtained by top down proteomics and immunoblotting. These references could further support the relevance of the novelty introduced in the present manuscript that could allow to get better insight into the mechanism of muscle metabolic adaptation in paraphysiological and pathological conditions.

Reviewer #3:

None

Rebuttal, Complex I is bypassed during high intensity exercise

Reviewer #1

The concept of a ETC complex 1 bypass is interesting during intense exercise is interesting, but I have some concerns with the metabolic model that the authors used. It is not described in detail and appears to be heavily “glucose” centric (see Figure 1A). I am not sure that the contribution of other carbon substrates, particularly fatty acids, is adequately considered in the model. This is particularly relevant to the proposed conclusion of a Complex 1 bypass, as much of the FADH₂ entering the ETC via Complex 2 originates from fatty acid β -oxidation. It is also not considered that during exercise circulating fatty acid levels increase, and can also provide an additional source of fuel for muscle.

We would first like to thank reviewer #1 for the insightful comments. We have now expanded the small-scale model to also account for fatty acid (octanoic acid) metabolism (Figure 1 and 2). We have also expanded the description of the model in the main text and added a more in-depth explanation in supplementary methods.

My major concern with the Complex 1 bypass model is where all the NADH is going. The major reduced equivalents produced is NADH, not FADH₂, and if Complex 1 is bypassed then muscle would need to deal with all this NADH. The authors propose that this NADH can be rerouted through the glycerol phosphate shuttle. Due to the amounts of NADH produced I find this hard to believe, and the authors do not provide any convincing evidence to support this possibility.

Whilst the calculated catalytic capacity of the glycerol phosphate shuttle shows that it is sufficient to support the flux of both cytosolic and mitochondrial NADH, we do agree that complex I bypass of mitochondrial NADH probably occurs through a different mechanism *in vivo*. We have moved our reasoning about this from the discussion to the results to make this distinction clear before proceeding with the analysis. It does not at all seem implausible that muscles deal with the electron transfer from mitochondrial NADH to ubiquinol by disengaging the proton pump in complex I. Different isoforms of complex I with lower proton stoichiometry exists in microbes¹, and a disengagement of the proton pumps was recently demonstrated experimentally in *Yarrowia lipolytica*². Whilst detailed knowledge about the mechanism of complex I bypass is interesting in its own right, it is not required to proceed with the constraint-based analysis.

The concept is proposed that bypassing Complex 1 is a biological adaptation of muscle to situations such as exercise. However, it is not clear what advantage this bypass of Complex 1 would have during exercise. The net effect would be a decreased ATP production.

The advantage of bypassing complex I is that it supports higher ATP synthesis rates/gram muscle, since the specific catalytic activity of complex I is very low. We have improved our description of complex I bypass to clarify this (e.g. Figure 1B).

The authors propose that glycogen use provides additional ATP (3 ATP/glucose) compared to glycolysis (2 ATP/glucose). This is actually only relevant in a net glycogen depletion scenario. The synthesis of glycogen requires the consumption of 1ATP/glucose).

We agree, and it is also reasonable to assume that glycogen is depleted during exercise³, we now explicitly mention this assumption in conjunction with the discussion about the ATP yield on glycogen.

The conclusion that metabolic changes during exercise result in mitochondrial uncoupling and fatty acid synthesis in peripheral tissue is speculative and not based on evidence or modeling.

With regards to mitochondrial uncoupling, the modeling shows that an increased rate of lactate clearance would increase the steady-state work rate. Mitochondrial uncoupling could increase this rate and we have added a supplementary figure exploring this possibility (Figure S7). We also compiled some support from

literature sources, however, more research is required to determine if this *in silico* phenomena may also occur *in vivo*.

With regards to fatty acid synthesis, labeling studies do indeed suggest that lactate is primarily oxidized⁴, we therefore prevent the model from synthesizing fat from lactate. A lactate buffering reaction with HCO_3 has been added in its place to explain the excess CO_2 , as we previously proposed in the discussion.

Figure 2 C predicts that fatty acids are minor contributors to metabolic flux. This is hard to believe and not consistent with actual metabolic measurements of fatty acid flux. Indeed, in Figure 1B, the RER values are consistent with a significant contribution of fatty acid oxidation to oxygen consumption.

The high energy content of palmitate (ATP/mol) can give an impression of low fluxes when presented alongside glycogen. We have updated the figure to better show the relative importance of fat at different work rates (Figure 4C). We moved the more detailed description of the fluxes to a supplementary figure (Figure S6) and supplementary data.

I have concerns that the experimental validation of the modeling was performed using a high resolved incremental exercise test on only one individual.

We now increased the number of subjects to a total of 5 with consistent results.

Reviewer #2

The observation made are novel and I believe the paper is of interest for the scientific community , not only in endurance training but particularly on paraphysiological and pathological conditions such as adaptation to hypoxia and muscle disorders, in which it is known that the metabolic adaptation is one of the primary events to cope with muscle impairment.

We would first like to thank reviewer #2 for the interesting perspectives and outlook. We have added a discussion about paraphysiological and pathological conditions.

However there is an aspect that is not well explained, even if I understand that this was not probably relevant for the Authors. There is not a proper description of the type of muscle utilized to calculate in 8 healthy subject the absolute amount of proteins. Muscles are characterized by different fiber composition and the variability among subjects could be quite high, I think that a brief description could be helpful to better appreciate the results.

We have now added more details about the proteomics data. It was collected from the thigh muscle (*vastus lateralis*) of sedentary adults (4 female and 4 male) with normal BMI (24.3).

In addition there are not reference of papers describing the metabolic rewiring revealed by semiquantitative data obtained by top down proteomics and immunoblotting. These references could further support the relevance of the novelty introduced in the present manuscript that could allow to get better insight into the mechanism of muscle metabolic adaptation in paraphysiological and pathological conditions.

We have now attempted to provide such a discussion based on a review of the field⁵, but suggestions for relevant papers would be much appreciated.

Reviewer #3

While the concept of glycerol phosphate shuttle is not a new idea (Gerbitz et al., 1996, Mráček et al., 2013), the activation of glycogenolysis and the role of glycerol phosphate shuttle at high exercise intensities are novel. Nevertheless, the evidences reported in support of their conclusion are weak. The approach proposed provides interesting ideas on the potential mechanisms associated to the regulation of the bioenergetics in working skeletal muscle. However, the prediction power of this model is greatly attenuated without convincing experimental validation. The absence of adequate comparisons between simulations and experimental data limits the possibility to use the model to make inferences on the regulatory mechanisms. Thus, these results are not convincing and there are several concerns related to the model validation which have been addressed in the specific comments below.

We would first like to thank reviewer #3 for the in-depth review. We have now expanded the background and discussion sections to clarify that we study constraints on metabolism and phenomena that can be inferred from the catalytic capacity of enzymes. We agree that constraint-based models have very limited explanatory power with regards to the regulatory mechanisms at play. We now discuss that these aspects fall outside the scope of this study. With regards to the validation, see specific answers below.

The glycerol phosphate shuttle allows mitochondria to use NADH without complex I while it can still function to contribute to the oxidative phosphorylation. Yet the authors suggest (see line 60) an absence of complex I. Therefore, it is not clear whether simulations suggest that complex I contribution to the ATP production is not relevant when the catalytic efficiency for ATP increase (Figure 1B). Complex I contribution should be reported.

We realize that it may have been ambiguous to present the optimal enzyme allocation (where complex I is absent at some catalytic rates) in the same figure as the simulated enzyme allocation (where complex I is always used up until maximum capacity). We now display these concepts in separate figures (Figure 1 and 2).

The strategy to obtain the results in Figure 1D also includes the minimization of glycogen uptake. This strategy should be adequately justified because glycogen is only one of the fuel sources for myocyte. The model accounts for fatty acid oxidation but the analysis was not performed for fatty acid although both fat and carbohydrate are utilized by mitochondria during exercise.

We have now expanded the small-scale model to also accommodate fatty acid metabolism (which was only available for the genome-scale model in the previous version). We have also added a justification for substrate minimization⁶.

(l. 86) Authors stated that there is a good agreement between muscle fiber respiration data and simulations reported in Figure 1E. The reason for this statement is unclear. When permeabilized fibers are stimulated by ADP in the presence of glutamate within the polarographic system, complex I is not bypassed. In contrast, simulated respiration rate was obtained bypassing complex I. Because simulated and experimental data appear obtained under different condition, a direct comparison does not seem to be meaningful. Also, pseudo code file does not have any information on how simulations in Figure 1E are obtained. The strategy to simulate the respiration rate in presence of glutamate or glutamate and succinate is unclear.

We now more clearly distinguish literature data⁷ from simulated data in the figure (Figure 2D). Our modeling predicts that 3 catalytic constraints (ETF, complex I and complex IV) come into effect in muscle fibers. We expect that the respiration rate with fat as substrate is constrained by ETF, the rate with complex I substrate (glutamate) is constrained by complex I and the rate with complex II substrate (succinate) is constrained by complex IV. This assumes that complex I bypass is not directly induced by the presence of complex I substrate. We now explicitly mention this assumption.

Model development is not sufficiently described. Authors stated that the multi tissue model used in their study is based on a previous model (Bordbar et al. 2011). The latest model was proposed to simulate alanine cycle, Cori cycle, and absorptive state which presented metabolic characteristics different from those during muscle contraction.

Physical exercise involves temporal changes in oxygen and substrates delivery in different tissue-organ systems. Passive and active transport processes as well as blood flow increase from 5 to 20L/min are affecting the metabolic fluxes at high exercise intensity.

We now describe the model development in greater detail in the main text and supplementary methods. It is assumed that the blood flow is adjusted to support the fluxes at steady state. We now mention this assumption.

Oxygen delivery (convection and diffusion) is reported to limit muscle oxidative metabolism at high metabolic rate (Grassi 2000 and Wagner 1996). The VO₂ kinetics experimental data simulated in Figure 2B are affected by metabolism, blood flow and diffusion. Blood flow and diffusion do not appear in the model proposed. The switch from fat to carbohydrate utilization has been associated to impairment of fatty acid delivery to muscle during muscle contraction (Romijn et al. 1993). It seems that the model proposed does not reflect the current knowledge on fatty acid delivery in a contracting muscle.

The discussion about the active constraints on VO₂max is still ongoing^{8,9} as we highlight in our introduction. To avoid taking a position, we constrained VO₂max to the experimentally observed level in each subject (a phenomenological constrain). The biological explanation for this constraint may very well be blood flow, diffusion, or pulmonary oxygen uptake. Impaired fat delivery may indeed be an active constraint in human as proposed by Romijn¹⁰, but subjects fed a high fat diet show increased fatty acid oxidation¹¹, which suggests that the constraint can be overcome by adaptation. Our analysis of the catalytic capacity of muscle fibers shows that fatty acid metabolism is constrained. This could, however, still be consistent with a delivery constraint, since any catalytic capacity exceeding the delivery capacity would be futile. However, it is still not completely clear to us why intramyocellular fatty acids could not compensate for delivery constraints. We now discuss this study and considerations in the main text.

Some information of the model (m1, m2, and m3) has been reported or can be derived by the pseudo code file. However, the main components of the m1 and m2 and how they are related to the model to simulate muscle fiber (Figure 1) are not clear.

We now explain the construction of the model in further detail in the main text and supplementary methods. Briefly, the active constraints identified in the muscle fiber (Figure 2C) were used to constrain the flux in m1 and m2.

The comparison between simulated and experimental pulmonary oxygen uptake and carbon dioxide release data obtained during an incremental exercise test is not adequate to validate the model proposed. Metabolic fluxes at cellular (Beard 2005, Wu et al., 2007) and whole-body level (Kim et al, 2007) are available but were not used for validation. In vivo metabolic fluxes measured below and above the ventilatory threshold were previously obtained from exercise protocol with constant or incremental work load and provide data at unsteady and steady-state conditions useful for validation.

We found the dynamic models by Beard¹², Wu¹³ and Kim¹⁴ very interesting. But it was not clear to us how to compare their reported fluxes with ours, see specific comments below. To clarify the distinction between constraint-based modeling and dynamic modeling, we have added a section to the introduction and discussion.

Beard¹² developed a computational model of OXPHOS in the isolated mitochondria, based on a large number of parameters from literature and by fitting parameter with experimental data. The only flux reported in this paper is the oxygen flux in response to the concentration of phosphate buffer, which is given with a unit normalized by the concentration of cytochrome A. It is unclear how to compare this flux with our simulated oxygen fluxes in muscle fibers, as constraint-based modeling does not consider metabolite concentrations. We do, however, compare our simulated oxygen fluxes with experimentally determined oxygen fluxes reported in literature⁷ (Figure 2D).

Wu¹³ used the model developed by Beard¹² to analyze NMR data of the human forearm muscle from literature. The only type of flux reported in this paper is the ATP synthesis rate, for which they assume a flux of 0.22 mmol/l cell water/s at 65% W_{max} , which corresponds to around 6 mmol/gdw/h at max. This is in agreement with our simulated fluxes at vO_2max of around 5.5 (Figure 2B). Interestingly, the study includes data from patients with complex I deficiency, and they are modeled by assuming that complex I did not pump protons and that its activity was reduced 1000-fold. We now consider the effects of pathological complex I deficiency in the discussion.

Kim¹⁴ developed a multi-scale model of the human body and used it to simulate the fluxes for a subject at rest and exercising at 150 W (corresponding to 60% of vO_2max) for an hour. The predicted steady-state fluxes at rest in different organs are reported for the subject. Literature values and predictions are also reported for the glucose production flux (up to 2 mmol/min) during exercise. Our model was not designed to study subjects at rest, a simple ATP hydrolysis reaction in peripheral tissue is used to represent basal metabolism. We do however simulate the optimal glucose flux at different durations (Figure 6B). At our lowest predicted rate (255 W), a glucose flux of 4.3 mmol/min is predicted, which is similar (around 30% higher) to the experimentally determined flux on a per watt basis 13 vs 17 $\mu\text{mol}/\text{min}/\text{W}$.

The strategy to estimate the model parameters for the results reported in figure 2 should be reported in the main text of the manuscript. For some parameters, their physiological and/or biochemical meaning in the codes are unclear. Pseudo code material does not provide enough details. Can these parameters be used to simulate other exercise protocols, or they are specifically designed to simulate the incremental exercise protocol? Can the results obtained with this fitting be generalized? Another concern on the results reported in figure 2 is related to the fact that complex I bypass was imposed at high ATP production rates. What if this condition is not imposed in the author's model? It is still possible that the model can simulate VO_2 and VCO_2 data with some adjustable parameters without the key assumption. Simulation design should be further developed and integrated with more experimental data to provide convincing evidence in support of the hypothesis proposed.

We now measure the exchange fluxes for a total of 5 subjects to provide additional experimental data. A non-linear least square algorithm was implemented to fit 3 parameters for the different subjects, the maintenance expenditure (for the intercept), a scaling factor for the enzyme concentration (linearly dependent with the total muscle mass) and the EFT constraint (required to fit the CO_2 data). As expected the model with complex I bypass describes the experimental data better than a model without (Supplementary Figure S4). We used the parameterized model to predict the maximum work rates at different durations (Figure 6C), which demonstrates that the model can be generalized to other conditions than incremental exercise. We have included more details about the parameter estimation in the main text.

In this work, simulations are mainly obtained with flux balance analysis which is based on the assumption that the system is at steady-state. The pulmonary oxygen uptake and carbo dioxide release during exercise (Figure 2B) are not obtained at steady-state. The model assumptions and simulation strategy should be further clarified and discussed

As the increments in work rate are small and each work rate is maintained for a long time, it is assumed that pulmonary exchange fluxes are at steady state. We now mention this assumption and we also consider the limitations and benefits with steady state modeling in the introduction and discussion.

In general, lactate kinetic results from the interplay among lactate transport, glycolytic metabolic fluxes associated to glycolysis and pyruvate oxidation. However, the simulated lactate and VO_2 kinetics seem unrelated to these key metabolic processes while there is no validation of these simulations with experimental data. Metabolic processes should be available from the computation simulations. There are various data available on human exercise (Brooks 1998) and approaches to study metabolic system with ordinary differential equations has been established (Korzeniewski et al., 2015, Wu et al., 2007, Cabrera et al., 1998).

The dynamic model of lactate simulates the production (constant flux) and clearance (Michaelis–Menten dynamics) of lactate, corresponding to the metabolic processes of anaerobic glycolysis and oxidation of lactate (pyruvate). This model serves to illustrate that a delayed response in lactate uptake, due to prior accumulation in a large volume, is sufficient to explain the slow component. We found the papers by Korzeniewski¹⁵, Wu¹³ and Cabrera¹⁶ very interesting, but they represent very complex dynamic models with dozens or even hundreds of equations and numerous parameters. Furthermore, the work rates investigated by Cabrera¹⁶ are well below those where the slow component phenomena occurs. In this case we think that a simple model (effectively 1 equation), serves our purposes better. We have now modified the model formulation to be able to compare the simulations with experimental data reported in literature¹⁷ (Figure 5).

The authors stated that mitochondrial uncoupling (l.222) was simulated at the level of single fiber and contracting muscle during exercise. In the material submitted, I could not find any relationship for energy-dissipating pathway used in the model to quantify proton leak. Thus, it is important to clarify whether the dissipation of the proton gradient is inferred directly or indirectly by the flux balance analysis. An increase in oxygen consumption rate does not provide enough evidence in support of mitochondrial uncoupling. Additional simulations should be reported in support of this statement.

The modeling shows that an increased rate of lactate clearance would increase the steady-state work rate. Mitochondrial uncoupling could increase this rate. We have now added a figure where we simulate uncoupling (Figure S7) and a section in the main text where it is discussed. Although we compile some support in favor of the phenomenon from literature sources, we think that more research is required to determine if this *in silico* phenomenon also occurs *in vivo*.

Substrate utilization results for palmitate reported in figure 2c are not consistent with the current knowledge in fuel utilization at different intensity of exercise. The contribution of fat and carbohydrate oxidation to ATP supply changes with power output. As exercise intensity (power output) increases, the energy provided from carbohydrate- and fat-derived fuel increase until power output reaches the crossover point. After this point, fat utilization decreases with an increase of exercise intensity. Crossover usually occurs at a power output close to 45% VO₂max (Brooks et al., 1994). While glycogen utilization is consistent with model simulation (Figure 2c), palmitate utilization is not.

The high energy content of palmitate (ATP/mol) can give an impression of low fluxes when presented alongside glycogen. We have updated the figure to better show the relative importance of fat at different work rates (Figure 4C). We moved the more detailed fluxes to supplementary figures and supplementary data.

Computational (Korzeniewski et al., 2015) and experimental studies (Connet & Sahlin 1996, Sutton et al. 1981) suggest that high exercise intensities cause cytosolic acidification leading to an inhibition of anaerobic glycolysis and an increase of oxidative phosphorylation. These evidences do not support the conclusion of the work which suggests that glycolytic flux contribution to the ATP production is increasing with exercise intensity. The authors should discuss this apparent contradiction between the results of their work and those reported in the literature.

The differences between our results and results in literature seem subtitle to us. We do predict a switch from glycolytic flux to lactate consumption in m1 at high work rates (Figure 4C). Our dynamic model predicts a switch from glycolytic flux to lactate oxidation when lactate concentrations increases, consistent with increased OXPHOS. The experiments by Sutton¹⁸, show that cytosolic acidification, e.g. through accumulation of lactate, ultimately lead to exhaustion. Our model is not particularly well suited to study these types of non-steady-state conditions, but we have added a discussion about the potential effects of acidification.

Currently, two main mechanisms are proposed for the activation of the ATP-supply system in contracting skeletal muscle: a) a parallel activation of metabolic blocks including glycolysis and complexes of the electron transport chain (Korzeniewski 1998 and 2003, Korzeniewski et al., 2015); b) an activation by negative feedback via ADP and inorganic

phosphate (Beard 2005, Wu et al., 2007). Regarding these works of recognized literary merit, the approach proposed in this paper suggests the activation of specific metabolic pathway although inferences are limited by the methodology. It would be beneficial to discuss the advantages and disadvantages in using genome scale models in comparison to the approach of computational metabolic models that use ordinary differential equation.

We agree that mechanistic inference about how metabolism is activated is not possible with the current constraint-based model. This is also not our objective. We are interested in phenomena that can be inferred from the catalytic capacity of enzymes under the assumption of optimal regulation. We have added a section about dynamic models and regulation in the introduction and have expanded our discussion about genome-scale modeling.

(l. 12) The authors stated that ATP synthesis is constrained by the specific activity of the metabolic enzymes. This is a vague concept. Literature on muscle physiology has several articles reporting the crucial role of the enzyme activities of the glycolytic and oxidative systems. Indeed, a reduced activity of complexes of the electron transport chain cause a reduced ATP production in different tissue-organ system.

We intended to convey that the flux (v) cannot exceed the catalytic capacity (v_{\max}). We have now reformulated the sentence.

(l. 81) Authors stated, "There is also a strong correlation (Pearson's correlation 0.7, $p=3.5e-05$, log transformed data) between the flux capacity and predicted fluxes". This statement is not clear. Please specify which fluxes are predicted and what it is the flux capacity. Physiology and biochemistry meaning of these fluxes are relevant.

We have now added a figure (Figure 2C) explaining the concept and displaying which pathway the fluxes belong to.

Figure 1E. The comparison between simulated and experimental data according to the statement in the results section should be reported.

We have now updated the figure to show both simulated and experimental data (Figure 2D).

(l. 96) It is not clear what model (muscle fiber vs. multi tissue) is used to simulate the data reported in figure 1F. Also, What's the fraction of fibers with complex I bypassed to obtain 1.28? And how is it estimated at low work rates.

The figure showed experimental data from literature¹⁹, we have now made it into a separate figure to more clearly distinguish between experiments and simulations (Figure 3).

References

1. Szenk, M., Dill, K. A. & de Graff, A. M. R. Why Do Fast-Growing Bacteria Enter Overflow Metabolism? Testing the Membrane Real Estate Hypothesis. *Cell Syst.* **5**, 95–104 (2017).
2. Cabrera-Orefice, A. *et al.* Locking loop movement in the ubiquinone pocket of complex I disengages the proton pumps. *Nat. Commun.* **9**, 4500 (2018).
3. Hearris, M. A., Hammond, K. M., Fell, J. M. & Morton, J. P. Regulation of Muscle Glycogen Metabolism during Exercise: Implications for Endurance Performance and Training Adaptations. *Nutrients* **10**, 298 (2018).
4. Brooks, G. A. Mammalian fuel utilization during sustained exercise. *Comp. Biochem. Physiol. Part B Biochem. Mol. Biol.* **120**, 89–107 (1998).
5. Gelfi, C., Vasso, M. & Cerretelli, P. Diversity of human skeletal muscle in health and disease: Contribution of proteomics. *J. Proteomics* **74**, 774–795 (2011).
6. Brooks, G. A. & Mercier, J. Balance of carbohydrate and lipid utilization during exercise: the ‘crossover’ concept. *J. Appl. Physiol.* **76**, 2253–2261 (1994).
7. Larsen, S. *et al.* Biomarkers of mitochondrial content in skeletal muscle of healthy young human subjects. *J. Physiol.* **590**, 3349–3360 (2012).
8. van der Zwaard, S. *et al.* Maximal oxygen uptake is proportional to muscle fiber oxidative capacity, from chronic heart failure patients to professional cyclists. *J. Appl. Physiol.* **121**, 636–645 (2016).
9. Cardinale, D. A. *et al.* Muscle mass and inspired oxygen influence oxygen extraction at maximal exercise: Role of mitochondrial oxygen affinity. *Acta Physiol.* **0**, e13110 (2018).
10. Romijn, J. A. *et al.* Regulation of endogenous fat and carbohydrate metabolism in relation to exercise intensity and duration. *Am. J. Physiol. Metab.* **265**, E380–E391 (1993).
11. Zajac, A. *et al.* The effects of a ketogenic diet on exercise metabolism and physical performance in off-road cyclists. *Nutrients* **6**, 2493–2508 (2014).
12. Beard, D. A. A Biophysical Model of the Mitochondrial Respiratory System and Oxidative Phosphorylation. *PLOS Comput. Biol.* **1**, e36 (2005).
13. Wu, F., Jeneson, J. A. L. & Beard, D. A. Oxidative ATP synthesis in skeletal muscle is controlled by substrate feedback. *Am. J. Physiol. Physiol.* **292**, C115–C124 (2007).
14. Kim, J., Saidel, G. M. & Cabrera, M. E. Multi-scale computational model of fuel homeostasis during exercise: Effect of hormonal control. *Ann. Biomed. Eng.* **35**, 69–90 (2007).
15. Korzeniewski, B. & Rossiter, H. B. Each-step activation of oxidative phosphorylation is necessary to explain muscle metabolic kinetic responses to exercise and recovery in humans. *J. Physiol.* **593**, 5255–5268 (2015).
16. Cabrera, M. E., Saidel, G. M. & Kalhan, S. C. Lactate metabolism during exercise: analysis by an integrative systems model. *Am. J. Physiol. Integr. Comp. Physiol.* **277**, R1522–R1536 (1999).
17. Sahlin, K., Sørensen, J. B., Gladden, L. B., Rossiter, H. B. & Pedersen, P. K. Prior heavy exercise eliminates $\dot{V}O_2$ slow component and reduces efficiency during submaximal exercise in humans. *J. Physiol.* **564**, 765–773 (2005).
18. Sutton, J. R., Jones, N. L. & Toews, C. J. Effect of pH on Muscle Glycolysis during Exercise. *Clin. Sci.* **61**, 331 LP–338 (1981).
19. Zoladz, J. A. *et al.* MYHC II content in the vastus lateralis m. quadriceps femoris is positively correlated with the magnitude of the non-linear increase in the $\dot{V}O_2$ / power output relationship in humans. *J. Physiol. Pharmacol.* **53**, 805–821 (2002).

Reviewer #1:

Remarks to the Author:

I still have some concerns regarding the simplicity of the proposed model. I am still concerned that the contribution of other carbon substrates, particularly fatty acids, is not adequately considered in the model. The authors now suggest that the model now accounts for octanoate metabolism. This is not a physiological fatty acid, compared to palmitate or oleic acid.

In my original review I raised concerns with the Complex 1 bypass model is where all the NADH is going. I still do not believe the authors provide any convincing evidence to support the role of the glycerol phosphate shuttle in this process.

The authors have still not provided convincing evidence as to what advantage this bypass of Complex 1 would have during exercise, with regard to ATP production.

My concern about ATP production from glycogenolysis has been adequately addressed.

I still believe that the metabolic changes during exercise result in mitochondrial uncoupling and fatty acid synthesis in peripheral tissue is speculative and not based on evidence or modeling.

I do not believe the authors have made a convincing argument that fatty acids are a minor contributor to metabolic flux.

My concern about experimental validation of the modeling being performed using a high resolved incremental exercise test on only one individual has been adequately addressed with the addition of more subjects.

Reviewer #2:

Remarks to the Author:

Thank you for considering the Reviewer's comments. However, since this is an important aspect not only under physiological conditions or exercising but it can have profound implications also in muscle diseases in which the lack of ATP leads to muscle waste and impairment in muscle cell renewal and apoptosis. In my opinion the cited review is too old I will suggest more recent specific articles on parapsychological (Capitanio et al. Sci. Rep. 2017) and pathological conditions (De Palma et al. J. of Prot. Res 2014, and Hughes M et al, Journal of Cachexia, Sarcopenia and Muscle, 2019). In line with these recent publications, it could be also useful if you can provide a comment on experimental findings and on the metabolic rewiring of the TCA cycle in the context of such conditions proposed.

Reviewer #3:

Summary

I appreciate the authors' efforts in addressing my previous concerns. Most of the concerns were addressed while more information is accessible in the revised version. I still have a few comments regarding some of the author responses included in the current version.

Point 4. The section that describes the simulation strategy for the results reported in Figure 2D is still incomplete. But from the caption of figure 2, I understand that you selectively choose the absence of complex I bypass to simulate the respiration rate with GM3 and the presence of complex I bypass for the simulation of the fiber respiration rate with GM3S and PGMSOC3 substrates. To appreciate the effect of complex I bypass, it would be meaningful to simulate fiber respiration rate in presence of PGMSOC3 and GMS3 without complex I bypass (blocking glycerol phosphate). My comment is related to the fact that even without glycerol phosphate, C-I and C-II activities should be enough to have a simulated respiration rate similar or greater than that observed experimentally with GM3S. Also, C-I, C-II and ETF activities should be enough to have a simulated respiration rate like that observed experimentally with PGMSOC3 substrates. Is your model excluded this possibility? The simulations suggested can be used to answer this question. Also, what about the simulation for GM3 with the effect of glycerol phosphate? These simulations allow you to compare the simulated respiration rates with and without complex I bypass for the same substrate utilization.

Complex I activity. The complex I activity reported in Table S1 and S2 is particularly important because it appears to be used for the simulations of this work. According to Table S1, the enzyme activity used in your simulations seems 6.6 U/mg. If the complex I activity is expressed in U/mg as it appears, it is 60-fold higher than that observed for isolated human skeletal muscle mitochondria (0.1 U/mg, Hoppel et al., 1987). It should be noted that this study and those you mentioned reported the activity normalized to the mass of mitochondrial protein although it can be also normalized to the homogenate skeletal muscle mass (Brooks et al, 2000). This aspect should be clarified because your results are based on protein content and this specific activity is referred to mitochondrial protein which is quite different from protein of tissue. Specifically, 1 g of skeletal muscle has approximately only 42 mg of mitochondrial protein (Brooks et al, 2000). It is acceptable that complex I activity varies with the enzyme assay protocol used, but the value used in your simulations appears much higher in comparison to those reported for human and rat skeletal muscle (Brooks et al, 2000, Hoppel et al., 1987). Thus, this low complex I activity (0.1-0.5 U/mg) value should indirectly support your hypothesis. Furthermore, can a 0.1-0.5 U/mg of complex I activity in your model, still sustains human exercise for metabolic rate under the ventilatory threshold? (see your figure 4C).

I believe reference # 4 in Table S1 and S2 used for complex I activity, it is not the correct one. In that paper, I could not find any data on Ovis aries. They work on rabbit and do not provide data on complex I.

References

Hoppel, C.L., Kerr, D.S., Dahms, B., Roessmann, U., 1987. Deficiency of the reduced nicotinamide adenine dinucleotide dehydrogenase component of complex I of mitochondrial electron transport. *J. Clin. Invest.* 80, 71–77.

Brooks H and Krahenbuhl S. Development of a New Assay for Complex I of the Respiratory Chain *Clinical Chemistry* 46:3 345–350 (2000).

Minor concerns

l. 38 sentence: “A number of potential limiting factors have been proposed, including the pulmonary uptake rate of oxygen,....”. Pulmonary uptake rate of oxygen is used to quantify VO_2max , it is not a limiting factor.

l. 107 sentence: “It reduces the P/O ratio of the cell, but also reduces the protein investment required to sustain oxidative phosphorylation”. Could you further explain it?

l. 188 sentence: “Based on the difference, the oxygen expenditure at high work rates is expected to”
What difference?

lines 150, 265, 363, and 767: Please check the use of the word “saturated”. Saturated with what? I would reconsider the use of the word saturated in some of these sentences.

l.255 Figure 3BA?

l. 291 sentence: “Since our multi-tissue model assumes that the lactate flux is at steady state, it factors in the slow component in its predictions.” Please revise this sentence.

Response to referees

Reviewer #1

I still have some concerns regarding the simplicity of the proposed model. I am still concerned that the contribution of other carbon substrates, particularly fatty acids, is not adequately considered in the model. The authors now suggest that the model now accounts for octanoate metabolism. This is not a physiological fatty acid, compared to palmitate or oleic acid.

It is correct that octanoate is not a physiological fatty acid compared to palmitate or oleic acid. It was chosen since it is the fatty acid used in the muscle fiber experiment (Larsen *et al.*, 2012) that is simulated in Figure 2D. Similarly, glutamate and succinate are not physiological substrates compared to glucose and glycogen. We now point this out in the main text (**line 169-171**). From a modeling perspective, the use of octanoate instead of palmitate has few implications since it involves the same enzymes and since palmitoyl-CoA is converted into octanoyl-CoA after a few cycles of beta oxidation (via myristoyl-CoA, lauroyl-CoA and decanoyl-CoA) while the output of acetyl-CoA, NADH and FAD is similar for each cycle. All of the enzyme mediated reactions in the model are listed in supplementary Table S1 if there are concerns about specific reactions.

In my original review I raised concerns with the Complex 1 bypass model is where all the NADH is going. I still do not believe the authors provide any convincing evidence to support the role of the glycerol phosphate shuttle in this process.

The NADH-capacity of Glycerol-3-phosphate dehydrogenase (GPD) is 2.1 mmol/gdw/h, based on its specific activity and enzyme concentration. This is sufficient to support the excess NADH flux in our simulations, 1.2 mmol/gdw/h. The simulations are only affected by biophysical constraints, e.g. when complex I reaches its capacity constraint at 0.88 mmol/gdw/h (Figure 2C) and the existence of alternative NADH sinks would not affect the results since the capacity of GPD is already sufficient under all simulated conditions. We now mention that the capacity of GPD is sufficient to support the flux of NADH in our simulations (**line 162-164**). Still it is possible that also other enzymes than GPD contribute to the NADH turnover *in vivo*, as we discuss on (**line 119-132**).

The authors have still not provided convincing evidence as to what advantage this bypass of Complex 1 would have during exercise, with regard to ATP production.

The main advantage of complex I bypass is that it enables increased ATP synthesis without invoking fermentation, which cannot be sustained for long. To synthesize equal amounts of ATP using full respiration would require a larger investment in proteins, with direct ATP costs for production and maintenance or opportunity costs in the form of reduced concentrations of other proteins if the total protein content is conserved. Complex I bypass thus enables a balance between ATP synthesis, protein allocation and substrate efficiency and constitutes a middle ground between the fully fermentative and the fully aerobic mode. We now explicitly state this (**line 106-112**) and also changed the caption of figure 1 to make it more explicit.

My concern about ATP production from glycogenolysis has been adequately addressed.

I still believe that the metabolic changes during exercise result in mitochondrial uncoupling and fatty acid synthesis in peripheral tissue is speculative and not based on evidence or modeling.

We do agree that fatty acid synthesis in peripheral tissues was inconsistent with labeling experiments from literature (Brooks, 1998). This option was therefore blocked already in the previous revision and does not appear in the revised version of the manuscript.

We do agree that uncoupling as a metabolic strategy was insufficiently described in the first version of the manuscript. We do, however, believe that our revision has addressed this. Clearance of lactate by uncoupling in peripheral tissues becomes an optimal strategy if the lactate synthesis flux is constrained due to lack of a sink. Our simulations show that the steady state work rate could increase if this option was available to the model (**Supplementary Figure S7B**). Labeling studies have indeed shown that most (55-70%) of the lactate is oxidized after exhausting exercise (Brooks, 1998). We have now added this reference to the main text (**line 376-377**).

I do not believe the authors have made a convincing argument that fatty acids are a minor contributor to metabolic flux.

We agree that there is marked contribution from fatty acid metabolism at low work rates, as evident from figure 4C. It is, however, well established that the oxidation of fatty acids have a maximum at intermediate work rates and decreases to a minor contribution at high work rates (Romijn *et al.*, 1993; Achten and Jeukendrup, 2003). This is in agreement with our gas exchange measurements, that show a RER>1 at high work rates, consistent with complete carbohydrate metabolism.

My concern about experimental validation of the modeling being performed using a high resolved incremental exercise test on only one individual has been adequately addressed with the addition of more subjects.

Reviewer #2

*Thank you for considering the Reviewer's comments. However, since this is an important aspect not only under physiological conditions or exercising but it can have profound implications also in muscle diseases in which the lack of ATP leads to muscle waste and impairment in muscle cell renewal and apoptosis. In my opinion the cited review is to hold I will suggest more recent specific articles on paraphysiological (Capitanio *et al. Sci. Rep.* 2017) and pathological conditions (De Palma *et al. J. of Prot. Res.* 2014, and Hughes M *et al, Journal of Cachexia, Sarcopenia and Muscle*, 2019). On line with this recent publications, it could be also useful if you can provide a comment on experimental findings and on the metabolic rewiring of the TCA cycle in the context of such conditions proposed.*

Rewiring of the TCA cycle and the use of alternative carbon sources, e.g. glutamine, is very interesting. With a constraint on the flux through complex I, the yield of ATP per NADH produced becomes important, and it is more than 20% higher when oxidizing glutamine to alanine, compared with oxidizing lactate to CO₂ (3.5 vs 2.85 ATP/NADH). An impaired ability to bypass complex I would be consistent with reduced oxidative capacity and increased production of complex I dependent H₂O₂ normalized to the oxygen consumption as seen in muscular dystrophy (Hughes M *et al, Journal of Cachexia, Sarcopenia and Muscle*, 2019). For patients with complex I deficiency, a metabolic bypass of complex I, using a membrane-permeable form of succinate, has been proposed as treatment, and shown effective in complex I deficient blood cells (Ehinger *et al.*, 2016). It is also interesting to see that mice that are exposed to hypoxia for 2 days have increased activity of complex I at the expense of glycolytic enzymes (Capitanio *et al. Sci. Rep.* 2017), potentially to make more efficient use of the limited oxygen resources under these conditions, whilst they after 10 days have reverted to the baseline, potentially due to restored oxygen availability through adjustment of the red blood cell count. The transient increase in Complex I activity stands in contrast to other studies, where chronic exposure to hypoxia increases the level of OXPHOS proteins. But this may possibly be a species-specific response.

We have now separated the paraphysiological and pathological conditions into separate paragraphs (**line 441-451 and 452-468**) to emphasize their independent importance. We have complemented the discussion with the more recent articles cited above. And we briefly mention the existence of other forms of metabolic rewiring.

Reviewer #3

I appreciate the authors' efforts in addressing my previous concerns. Most of the concerns were addressed while more information is accessible in the revised version. I still have a few comments regarding some of the author responses included in the current version.

Point 4. The section that describes the simulation strategy for the results reported in Figure 2D is still incomplete. But from the caption of figure 2, I understand that you selectively choose the absence of complex I bypass to simulate the respiration rate with GM3 and the presence of complex I bypass for the simulation of the fiber respiration rate with GM3S and PGMSOC3 substrates. To appreciate the effect of complex I bypass, it would be meaningful to simulate fiber respiration rate in presence of PGMSOC3 and GMS3 without complex I bypass (blocking glycerol phosphate). My comment is related to the fact that even without glycerol phosphate, C-I and C-II activities should be enough to have a simulated respiration rate similar or greater than that observed experimentally with GM3S. Also, C-I, C-II and ETF activities should be enough to have a simulated respiration rate like that observed experimentally with PGMSOC3 substrates. Is your model excluded this possibility? The simulations suggested can be used to answer this question. Also, what about the simulation for GM3 with the effect of glycerol phosphate? These simulations allow you to compare the simulated respiration rates with and without complex I bypass for the same substrate utilization.

We have modified the simulation strategy to accommodate these requests with similar results. We now simulate GM3 by providing glutamate as substrate to the model, assuming that it is converted to aspartate as seen in isolated mitochondria (Kovačević, 1971). We simulate GM3S by providing both glutamate and succinate as substrate, assuming that succinate is converted to fumarate. We also show the effect of complex I bypass for all of the simulations. The effect of complex I bypass is only noteworthy in the simulation that depend on complex I substrate, as could be expected. For simulations with octanoic acid the constraint from ETF cannot be circumvented, and the results are therefore independent of the complex I constraint. For succinate the capacity of complex II is almost sufficient to reach the CIV constraint independent of complex I bypass. We have updated Figure 2D with the new results and also updated the method section and main text as well as supplementary methods to reflect these changes (**Line 165-182 and 523-531**).

Complex I activity. The complex I activity reported in Table S1 and S2 is particularly important because it appears to be used for the simulations of this work. According to Table S1, the enzyme activity used in your simulations seems 6.6 U/mg. If the complex I activity is expressed in U/mg as it appears, it is 60-fold higher than that observed for isolated human skeletal muscle mitochondria (0.1 U/mg, Hoppel et al., 1987). It should be noted that this study and those you mentioned reported the activity normalized to the mass of mitochondrial protein although it can be also normalized to the homogenate skeletal muscle mass (Brooks et al, 2000). This aspect should be clarified because your results are based on protein content and this specific activity is referred to mitochondrial protein which is quite different from protein of tissue. Specifically, 1 g of skeletal muscle has approximately only 42 mg of mitochondrial protein (Brooks et al, 2000). It is acceptable that complex I activity varies with the enzyme assay protocol used, but the value used in your simulations appears much higher in comparison to those reported for human and rat skeletal muscle (Brooks et al, 2000, Hoppel et al., 1987). Thus, this low complex I activity (0.1-0.5 U/mg) value should indirectly support your hypothesis. Furthermore, can a 0.1-0.5 U/mg of complex I activity in your model, still sustains human exercise for metabolic rate under the ventilatory threshold? (see your figure 4C).

This value is indeed important, and it is reported in the unit $\mu\text{mol}/\text{min}/\text{mg}$ purified protein. This usage of the term “specific activity” is consistent with BRENDA's use, although others do exist, e.g. U/mg muscle mass or U/mg mitochondrial protein. This has now been clarified in the main text (**line 86**) and the legends of Table S1 and S2. The specific activity value is used together with proteomics data to calculate the value in the unit U/mg muscle mass, using the relation $v_{\text{max}} = [\text{specific activity}] * [\text{enzyme concentration}]$. With a concentration of Complex I of around 0.5% of the total muscle protein (Figure 2A) the activity in the unit U/mg muscle mass becomes around 0.03, and correcting for the

concentration of mitochondria by a factor 24 (1000/42 mg), it corresponds to a value of around 0.7 U/mg mitochondria, which is similar, albeit slightly higher than the suggested values. A specific activity of 0.1 U/mg mitochondria is thus not expected to be sufficient to sustain human exercise up until the ventilatory threshold.

I believe reference # 4 in Table S1 and S2 used for complex I activity, it is not the correct one. In that paper, I could not find any data on Ovis aries. They work on rabbit and do not provide data on complex I.

The correct reference is “Letts JA, Degliesposti G, Fiedorczuk K, Skehel M, Sazanov LA. Purification of Ovine Respiratory Complex I Results in a Highly Active and Stable Preparation. J Biol Chem . 2016;291: 24657–24675.”. This has now been corrected. We reexamined all other references in Table S2 and found them to be correct.

I. 38 sentence: “A number of potential limiting factors have been proposed, including the pulmonary uptake rate of oxygen,....”. Pulmonary uptake rate of oxygen is used to quantify VO2max, it is not a limiting factor.

We had the uptake capacity of oxygen by the lungs in mind. We have now reformulated the sentence.

I. 107 sentence: “It reduces the P/O ratio of the cell, but also reduces the protein investment required to sustain oxidative phosphorylation”. Could you further explain it?

We have now reformulated the paragraph with more straight forward language.

I. 188 sentence: “Based on the difference, the oxygen expenditure at high work rates is expected to” What difference?

We are comparing the expected oxygen expenditure per watt produced at low and high work rates assuming complete bypass of complex I at high work rates. The sentence has now been reformulated.

lines 150, 265, 363, and 767: Please check the use of the word “saturated”. Saturated with what? I would reconsider the use of the word saturated in some of these sentences.

For lines 150 and 265 we were aiming at “attained its maximum capacity”. For line 363 the intent was that “other options were exhausted” and for line 767 “at full capacity”. The text has now been updated.

I.255 Figure 3BA?

“Figure 3A”. It has now been corrected.

I. 291 sentence: “Since our multi-tissue model assumes that the lactate flux is at steady state, it factors in the slow component in its predictions.” Please revise this sentence.

We wanted to communicate that our model operates under the steady state assumption, which in the case of the slow component means that the predicted oxygen expenditure is expected to be higher than the observed if a subject has not yet reached steady state. We have tried to revise the sentence to make this clearer.

References

- Achten, J. and Jeukendrup, A. E. (2003) 'Maximal Fat Oxidation During Exercise in Trained Men', *Int J Sports Med*, 24(08), pp. 603–608. doi: 10.1055/s-2003-43265.
- Brooks, G. A. (1998) 'Mammalian fuel utilization during sustained exercise', *Comparative Biochemistry and Physiology Part B: Biochemistry and Molecular Biology*, 120(1), pp. 89–107. doi: [https://doi.org/10.1016/S0305-0491\(98\)00025-X](https://doi.org/10.1016/S0305-0491(98)00025-X).
- Ehinger, J. K. *et al.* (2016) 'Cell-permeable succinate prodrugs bypass mitochondrial complex I deficiency', *Nature communications*. Nature Publishing Group, 7, p. 12317. doi: 10.1038/ncomms12317.
- Kovačević, Z. (1971) 'The pathway of glutamine and glutamate oxidation in isolated mitochondria from mammalian cells', *Biochemical Journal*, 125(3), pp. 757 LP – 763.
- Larsen, S. *et al.* (2012) 'Biomarkers of mitochondrial content in skeletal muscle of healthy young human subjects', *Journal of Physiology*, 590(14), pp. 3349–3360. doi: 10.1113/jphysiol.2012.230185.
- Romijn, J. A. *et al.* (1993) 'Regulation of endogenous fat and carbohydrate metabolism in relation to exercise intensity and duration', *American Journal of Physiology-Endocrinology and Metabolism*. American Physiological Society, 265(3), pp. E380–E391. doi: 10.1152/ajpendo.1993.265.3.E380.

Reviewers' Comments:

Reviewer #1:

Remarks to the Author:

Despite the two revisions to the paper, I still have concerns regarding the simplicity of the proposed model. I am still concerned that the contribution of other carbon substrates, particularly fatty acids, is not adequately considered in the model. The authors now suggest that the model now accounts for octanoate metabolism. The suggestion is that octanoate is metabolized in a manner similar to physiological fatty acids. This is incorrect, as octanoate can completely bypass the mitochondrial CPT1/CPT2 pathway. In addition, suggesting that octanoate is appropriate in the modelling studies because it was used in the muscle fiber experiments is not an adequate justification. I am sure that octanoate was used in the muscle fiber experiments because it was easier to use than physiological fatty acids (because of the need to bind them to albumin).

In my original review I raised concerns with the Complex 1 bypass model is where all the NADH is going. I still do not believe the authors provide any convincing evidence to support the role of the glycerol phosphate shuttle in this process. While it is now suggested that glycerol-3-phosphate dehydrogenase (GPD) has enough capacity to deal with the mitochondrial NADH, I find this concept is confusing. Normally, cytoplasmic GPD functions to shuttle reduced equivalents into the mitochondria. The mitochondrial GPD normally reduces FAD to FADH₂. For the mitochondrial GPD to reverse its activity and to take mitochondrial NADH to produce NAD⁺, and then have the glycerol shuttle out of the mitochondrial where cytosolic GPD would reverse its activity and produce NADH does not seem logical. The authors provide no evidence that this pathway is functioning.

The authors have still not provided convincing evidence as to what advantage the bypass of Complex 1 would have during exercise, with regard to ATP production. The argument that it prevents fermentation and that synthesizing equal amounts of ATP using full respiration would require a larger investment of proteins is not at all convincing. Bypassing Complex 1 will decrease the efficiency of synthesizing ATP.

My concern that the metabolic changes during exercise result in mitochondrial uncoupling and fatty acid synthesis in peripheral tissue is speculative and not based on evidence or modeling has now been adequately addressed in the second revision.

I still do not believe the authors have made a convincing argument that fatty acids are a minor contributor to metabolic flux. In fact, the two papers now cited by the authors actually show that at 50-75% of VO₂ max that fatty acid oxidation provides over 30% of whole body energy production. In fact, the Romijn et al, 1993 paper shows that intense exercise increases endogenous muscle fatty acid mobilization for oxidation.

Reviewer #2:

Remarks to the Author:

The Author provided comments and references on parapsychological and pathological conditions in which rewiring was observed in muscle tissue. However, I understand that under such conditions a specific study will be needed to model the contribution of complex I dysregulation to muscle metabolic adaptation. It is a very interesting paper that highlight the plasticity of the muscle to adapt to energy requirements.

Reviewer #3:

Remarks to the Author:

Authors properly addressed all my concerns.

Response to referees

Reviewer #1

Despite the two revisions to the paper, I still have concerns regarding the simplicity of the proposed model. I am still concerned that the contribution of other carbon substrates, particularly fatty acids, is not adequately considered in the model. The authors now suggest that the model now accounts for octanoate metabolism. The suggestion is that octanoate is metabolized in a manner similar to physiological fatty acids. This is incorrect, as octanoate can completely bypass the mitochondrial CPT1/CPT2 pathway. In addition, suggesting that octanoate is appropriate in the modelling studies because it was used in the muscle fiber experiments is not an adequate justification. I am sure that octanoate was used in the muscle fiber experiments because it was easier to use than physiological fatty acids (because of the need to bind them to albumin).

The reviewer is correct that transport of palmitate relies on the carnitine shuttle. These extra enzymatic steps should make palmitate a less favorable substrate than octanoate with regards to catalytic efficiency. However, also other differences affect the efficiency, e.g. the ATP cost of fatty acid activation is twice as costly per c-mol for octanoate than palmitate due to differences in chain length (2/8 vs 2/16). We have therefore expanded the model (reactions in **Supplementary Table S2**) to include oxidation of palmitate. We find that palmitate is around 5% more catalytic- and substrate efficient compared with octanoate. We have now updated the figure (**Figure 1**) with palmitate as substrate. This had no qualitative effect on the order of metabolic modes.

In my original review I raised concerns with the Complex 1 bypass model is where all the NADH is going. I still do not believe the authors provide any convincing evidence to support the role of the glycerol phosphate shuttle in this process. While it is now suggested that glycerol-3-phosphate dehydrogenase (GPD) has enough capacity to deal with the mitochondrial NADH, I find this concept is confusing. Normally, cytoplasmic GPD functions to shuttle reduced equivalents into the mitochondria. The mitochondrial GPD normally reduces FAD to FADH₂. For the mitochondrial GPD to reverse its activity and to take mitochondrial NADH to produce NAD⁺, and then have the glycerol shuttle out of the mitochondrial where cytosolic GPD would reverse its activity and produce NADH does not seem logical. The authors provide no evidence that this pathway is functioning.

We agree with the reviewer that reversing mitochondrial GPD would not be logical. Aside from the undesirable effects on NADH it would also be physically impossible as it faces the IMS side of the mitochondrial membrane. This is not the route predicted by the model. In the simulation, electrons are transported from mitochondrial NADH to cytosolic NADH via the malate-aspartate shuttle operating in reverse; the glycerol phosphate shuttle then utilizes the cytosolic NADH to feed the electron transport chain via ubiquinol. We have updated Figure 1 and the main text (**lines 90-93** and **124-128**) to more clearly convey this (the simulated fluxes are reported in **Supplementary data 2**). While GPD does indeed appear to have sufficient capacity to support this flux, it seems like an implausible metabolic route *in vivo*. We therefore speculate that an uncoupled form of complex I, as observed in microbes, may be more plausible, and we also find that this route would be preferred by the model if it could occur (**Supplementary Figure S1**). While we are unable to prove by which mechanism, we find that bypassing complex I would result in increased catalytic efficiency (due to the low efficiency of complex I), and we show that the observed fluxes on cellular level as well as whole-body level are consistent with this. We have now updated the discussion section (**lines 412-425**) to make this clearer.

The authors have still not provided convincing evidence as to what advantage the bypass of Complex 1 would have during exercise, with regard to ATP production. The argument that it prevents fermentation and that synthesizing equal amounts of ATP using full respiration would require a larger investment of proteins is not at all convincing. Bypassing Complex 1 will decrease the efficiency of synthesizing ATP.

A distinction must be made between catalytic efficiency (or catalytic capacity) and substrate efficiency (or yield) of a metabolic pathway (Molenaar *et al.*, 2009). The catalytic capacity of a pathway denotes the amount of product (e.g. ATP) that can be produced per gram of protein per unit of time. Substrate efficiency denotes the amount of product that can be produced per mol of substrate that is feed in to the pathway. Full respiration has higher substrate efficiency than fermentation (> 30 mol ATP/mol glucose compared with 2). However, full respiration has lower catalytic capacity than fermentation (~200 vs 400 mmol ATP/g protein/h). This means that a muscle specialized in fermentation can produce ATP at twice the rate of a muscle specialized in respiration, assuming that the same amount of protein is invested into metabolic enzymes in both muscles. However, the fermentative strategy is not sustainable, as substrate reserves rapidly decline and lactate accumulates. Complex I bypass provides a compromise between these strategies with intermediate substrate efficiency (>20 mol ATP/mol glucose) and intermediate catalytic capacity (~250 mmol ATP/g protein/h). It thus allows ATP synthesis at a higher rate than the fully fermentative strategy, whilst still being sustainable through a comparably high substrate efficiency and absence of fermentation products. The evidence for this is provided in **Figure 1B**. We have now updated the main text (**lines 105-123**) to make this clearer.

My concern that the metabolic changes during exercise result in mitochondrial uncoupling and fatty acid synthesis in peripheral tissue is speculative and not based on evidence or modeling has now been adequately addressed in the second revision.

*I still do not believe the authors have made a convincing argument that fatty acids are a minor contributor to metabolic flux. In fact, the two papers now cited by the authors actually show that at 50-75% of VO₂ max that fatty acid oxidation provides over 30% of whole body energy production. In fact, the Romijin *et al*, 1993 paper shows that intense exercise increases endogenous muscle fatty acid mobilization for oxidation.*

We do not wish to claim that fatty acids are minor contributors to the metabolic flux in general. At 75% of VO₂ max the model predicts a relative fatty acid utilization of around 25-30% (as shown in **Figure 4C**) in agreement with literature sources, e.g. Romijin *et al* observes the highest whole-body fatty acid oxidation at 65% of VO₂ max and levels are markedly lower at 85% of VO₂ max. However, at the highest work rates, our gas exchange measurements show a RER≥1 (**Figure 4B**), which is consistent with complete carbohydrate metabolism.

Reviewer #2

The Author provided comments and references on parapsiological and pathological conditions in which rewiring was observed in muscle tissue. However, I understand that under such conditions a specific study will be needed to model the contribution of complex I dysregulation to muscle metabolic adaptation. It is a very interesting paper that highlight the plasticity of the muscle to adapt to energy requirements.

Reviewer #3

Authors properly addressed all my concerns.

References

Molenaar, D. *et al.* (2009) 'Shifts in growth strategies reflect tradeoffs in cellular economics.', 2, 5, p. 323. doi: 10.1038/msb.2009.82.

Reviewers' Comments:

Reviewer #1:

Remarks to the Author:

none